# Flipped Classroom: Aligning Teacher Attention with Student in Generalized Category Discovery

**Haonan Lin**[1][†]    **Wenbin An**[2][†]    **Jiahao Wang**[1]    **Yan Chen**[1][*]    **Feng Tian**[1][*]

**Mengmeng Wang**[3,4]    **Guang Dai**[4]    **Qianying Wang**[5]    **Jingdong Wang**[6]

[1] School of Comp. Science & Technology, MOEKLINNS Lab, Xi'an Jiaotong University
[2] School of Auto. Science & Engineering, MOEKLINNS Lab, Xi'an Jiaotong University
[3] College of Comp. Science & Technology, Zhejiang University of Technology
[4] SGIT AI Lab, State Grid Corporation of China
[5] Lenovo Research    [6] Baidu Inc

## Abstract

Recent advancements have shown promise in applying traditional Semi-Supervised Learning strategies to the task of Generalized Category Discovery (GCD). Typically, this involves a teacher-student framework in which the teacher imparts knowledge to the student to classify categories, even in the absence of explicit labels. Nevertheless, GCD presents unique challenges, particularly the absence of priors for new classes, which can lead to the teacher's misguidance and unsynchronized learning with the student, culminating in suboptimal outcomes. In our work, we delve into why traditional teacher-student designs falter in open-world generalized category discovery as compared to their success in closed-world semi-supervised learning. We identify inconsistent pattern learning across attention layers as the crux of this issue and introduce FlipClass—a method that dynamically updates the teacher to align with the student's attention, instead of maintaining a static teacher reference. Our teacher-student attention alignment strategy refines the teacher's focus based on student feedback from an energy perspective, promoting consistent pattern recognition and synchronized learning across old and new classes. Extensive experiments on a spectrum of benchmarks affirm that FlipClass significantly surpasses contemporary GCD methods, establishing new standards for the field.

## 1 Introduction

Teacher-Student architecture has proved its effectiveness in Semi-Supervised Learning (SSL) [66, 32, 53], which aims to take advantage of a large collection of unlabeled data, reducing the expensive costs of annotation [57, 90, 89] . Previous approaches tend to model $p(\text{student}|\text{teacher})$, where the teacher typically acts as a fixed point of reference, providing a form of "supervision prior" to guide the student [25, 42, 60]. This supervision comes from labeled data and is asymmetrical: while the teacher has robust prior knowledge and provides a stable learning signal, the student's knowledge is incomplete and evolving. The student learns from both the teacher's supervision and the data it is exposed to, trying to emulate the teacher by aligning its predictions with those of the teacher.

---

[*]Corresponding Authors.
[†]Equal contribution.
This work was partly completed during the internship at SGIT AI Lab, State Grid Corporation of China.

38th Conference on Neural Information Processing Systems (NeurIPS 2024).

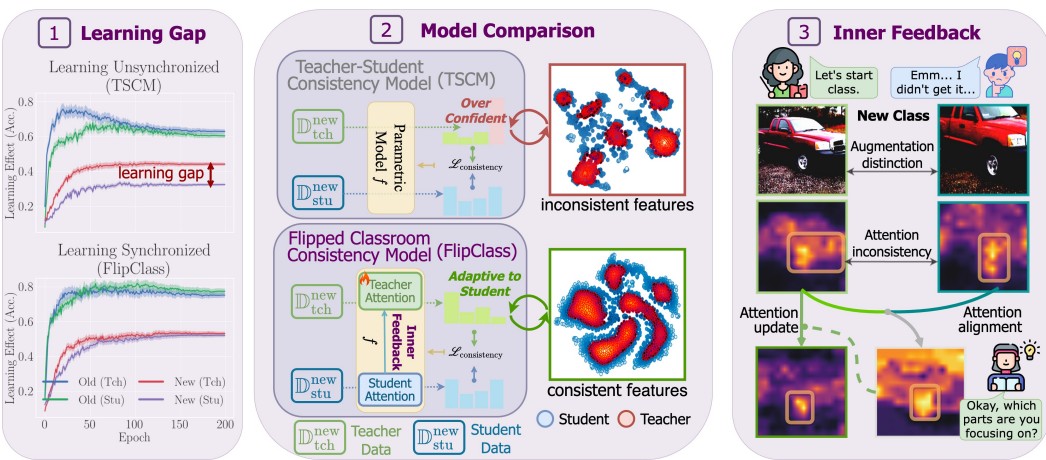

Figure 1: Left: Learning effects of traditional Teacher-Student Consistency Model (TSCM, *e.g.*, SimGCD [75]) and our Flipped Classroom Consistency Model (*FlipClass*) on Stanford Cars [33]. Middle: Model comparison between TSCM and our FlipClass, where $\mathbb{D}^{new}$ refers to data belonging to new classes. Right: Illustration of the inner feedback mechanism in FlipClass, where teacher attention is adapted to the student, leading to the alignment of attention.

Teacher-Student designs traditionally rely on a closed-world assumption, where it is expected that the teachers have supervision prior to all classes they will face while instructing students [61, 3]. However, real-world applications often involve dynamic and open environments, where instances belonging to new classes may appear [84, 30, 31, 41]. In such cases, discovering novelties could enable models to adapt new information and evolve continually as biological systems do [14, 49]. Recently, Generalized Category Discovery (GCD) [69] stands out by challenging models to categorize unlabeled data containing both old and new classes using only partial labels for training. Although recent GCD methods have adapted closed-world Teacher-Student strategies with notable success [75, 50], the transition is not seamless and presents several challenges.

**Challenge I: Learning gap.** Fig. 1 top left illustrates the learning evolution of student and teacher. The previous teacher-student models result in unsynchronized learning and a significant learning gap in new classes. The ideal learning dynamic between the teacher and student should be cohesive, which requires teaching students in accordance with their aptitude.

**Challenge II: Discrepancies in features.** The learning gap arises from the teacher's fast pace, leading to large discrepancies in representations between weakly-augmented data (teacher) and strongly-augmented data (student), especially for new classes (Fig. 1 middle). This causes significant prediction differences, making consistency loss optimization difficult and hindering effective student learning. Over time, the iterative learning process exacerbates this misalignment.

**Challenge III: Attention inconsistency.** Inadequate supervision for new classes leads to inaccurate instructions from the teacher, causing the student to focus on different parts than the teacher (Fig. 1 right). Imagine a classroom transitioning to a new subject. Without proper guidance, students' attention diverges from the teacher's, resulting in confusion and ineffective learning.

To sum up, the challenges of previous teacher-student models arise from inadequate supervision on new classes and the gap between weakly and strongly augmented data. This results in attention inconsistency (Chall. III), which leads to discrepancies in predictions and representations (Chall. II), ultimately causing a significant performance gap (Chall. I). Addressing these challenges requires developing teacher-student dynamics that align the evolving knowledge of both teacher and student (Fig. 1, bottom left). These findings lay the foundation for our approach, *FlipClass*, which models the teacher's posterior $p(\text{teacher}|\text{student})$ from the energy perspective of attention, building an adaptive teacher to bridge the learning gap. *FlipClass* offers a plug-and-play solution to foster an interactive learning environment where the student can influence the teacher's guidance in real-time, allowing teachers to tailor their instructions based on students' current attention [7, 1]. By aligning attention, *FlipClass* ensures that the learning pace of the teacher and student is in sync, leading to improvement on both old and new classes. Our contributions are summarized as:

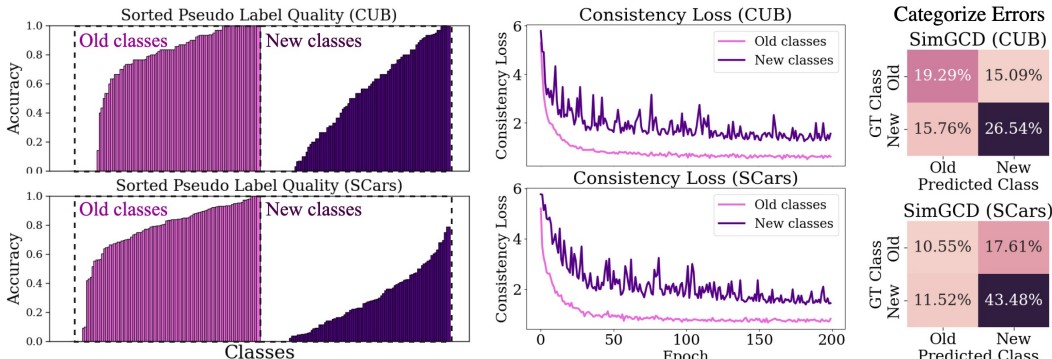

Figure 2: Exploring prior gaps between SSL and GCD on SCars and CUB datasets. Left: Accuracy of sorted pseudo labels for old and new classes. Middle: Consistency loss trends over epochs, illustrating challenges in optimization and slower convergence for new classes. Right: Categorize errors [75], where "True Old" refers to predicting an 'Old' class sample to another 'Old' class, while 'False Old" indicates predicting an 'Old' class sample as some 'New' class.

**(1) Empirical analysis**: We highlight the challenge of applying the closed-world Teacher-Student paradigm to the open-world scenario of GCD. Our in-depth analysis identifies attention misalignment between teacher and student as the key issue hindering synchronized learning between them.

**(2) Methodology**: Based on these analyses, we propose a flexible and effective method, *FlipClass*, which enables teachers to adapt and respond to student feedback to synchronize their learning progress, thereby leading to an overall improvement in teaching outcomes.

**(3) Superiority**: *FlipClass* consistently outperforms state-of-the-art generalized category discovery methods on both coarse-grained and fine-grained datasets.

## 2 Background

We first provide some background on semi-supervised learning and generalized category discovery to better contextualize our analysis. Let us consider a data set $\mathbb{D}$ consisting of labeled data $\mathbb{D}_L = \{(\mathbf{x}_i^\ell, y_i)\}_{i=1}^{N_L}$ from $|\mathbb{C}_K|$ old classes and unlabeled data $\mathbb{D}_U = \{\mathbf{x}_j^u\}_{j=1}^{N_U}$ that may contain instances from both old classes $\mathbb{C}_K$ and new classes $\mathbb{C}_N$, with $\mathbb{C} = \mathbb{C}_K \cup \mathbb{C}_N$. For a data instance $\mathbf{x}$, let $p_{\mathrm{m}}(y \mid f_\theta(\mathbf{x}))$ denote the predicted class distribution produced by the model $f$.

### 2.1 Integrating SSL Techniques into a Consistency Loss Framework

In semi-supervised learning (SSL), the goal is to enhance model performance by leveraging unlabeled data, traditionally drawn from the same class spectrum as the labeled data [90]. The goal of SSL can be formalized by integrating three fundamental techniques: *(1) Consistency regularization* ensures that the model outputs consistent predictions for augmented versions of the same instance. This technique utilizes different transformations to test the robustness of the model's predictions, promoting stability across variations in the input data [8, 59, 36]. *(2) Pseudo-labeling* utilizes the model to generate artificial labels for unlabeled data by adopting "hard" labels (that is, the argmax of the model output) and keeping only the labels where the highest class probability exceeds a predefined threshold [62, 47, 57, 37, 23]. *(3) Teacher-Student model* incorporates a structured learning relationship where the **teacher** model, typically trained on weakly-augmented instances, generates high-quality pseudo labels. These pseudo labels are then used to guide the training of the **student** model, which processes strongly-augmented instances. This approach helps improve the generalization capabilities of the student model by learning from the refined knowledge and stable supervision signals provided by the teacher [66, 43, 77, 12, 53]. Several methods integrate some of these techniques and achieve advanced performance in SSL [63, 83, 78, 87]. We unify these SSL techniques into one consistency loss:

$$\mathcal{L}_{\mathrm{cons}} = \frac{1}{|\mathbb{D}_U|} \sum_{\mathbf{x} \in \mathbb{D}_U} H\Big(p_{\mathrm{m}}\big(y \mid f_\theta(\alpha(\mathbf{x}))\big), p_{\mathrm{m}}\big(y \mid f_\theta(\mathcal{A}(\mathbf{x}))\big)\Big), \tag{1}$$

where cross-entropy $H(\cdot, \cdot)$ measures *consistency for regularization*, while the prediction $p_{\mathrm{m}}(y \mid f_{\boldsymbol{\theta}}(\mathbf{x}))$ serves as a *pseudo label*. This setup captures a *teacher-student dynamic*, where $\alpha(\mathbf{x})$ and $\mathcal{A}(\mathbf{x})$ represent the **teacher** (weakly-augmented) and **student** (strongly-augmented) instances, respectively.

## 2.2 Class Prior Gap between SSL and GCD

The GCD task pushes the boundaries of SSL by questioning the closed-world assumption that all classes in the unlabeled dataset $\mathbb{D}_U$ are previously known [69]. Instead, GCD incorporates new classes $\mathbb{C}_N$ into the unlabeled dataset, demanding that the model learn to recognize and then correctly classify them [4, 21, 80, 5]. In this open-world setting, SSL methods face obstacles with new classes due to the lack of supervision [24, 56], resulting in significantly lower quality of pseudo-labels for these new classes than for the old ones (Fig. 2 left). This gap exacerbates the complexity of optimizing the consistency loss Eq. 1 for new classes, leading to learning instability and slow convergence (Fig. 2, middle). Such optimization issues lead to severe prediction bias, resulting in new classes' performance lagging behind that of old classes (Fig. 2, right), underlining the limitations of existing SSL techniques in GCD scenarios. More empirical analysis can be found in *Appendix* B.2.

## 3 How Consistency Loss Goes Awry: Unraveling the Pitfalls

Acknowledging challenges presented by the absence of prior knowledge for new classes in traditional semi-supervised learning (SSL) methods is the first step toward addressing the complexities of open-world tasks. We further identify that the 'prior gap' manifests as issues in learning synchronization and representation discrepancy (Sec. 3.1). Our analysis targets the minimization of the energy function between teacher and student representations to bridge the 'prior gap'. We find that aligning their attention on similar patterns reduces energy, indicating effective alignment and learning (Sec. 3.2). This key understanding paves the way for the development of our proposed methods, aiming to synchronize teacher-student attentions for improved model learning dynamics (Sec. 4).

## 3.1 What to Bridge the Class Prior

The challenge of optimizing consistency loss leads to a learning gap between the student and the teacher, particularly evident when dealing with new classes (Fig. 1 top left). This gap causes the student to plateau, as it cannot keep pace with the teacher's more advanced understanding, which in turn restricts the teacher's progress in new classes Moreover, this learning gap also manifests itself in the divergent representations between teacher and student (Fig. 1 middle), specifically for new classes. Based on these observations, we revisit consistency loss (Eq. 1) in the closed-world setting.

**Insight 3.1.** *The large discrepancy between $f_{\boldsymbol{\theta}}(\alpha(x))$ and $f_{\boldsymbol{\theta}}(\mathcal{A}(x))$ complicates maintaining consistency across the model predictions. To narrow this divide, an intuitive idea is to align $f_{\boldsymbol{\theta}}(\alpha(\mathbf{x}))$ more closely with $f_{\boldsymbol{\theta}}(\mathcal{A}(\mathbf{x}))$, simplifying the optimization of $\mathcal{L}_{cons}$:*

$$\mathcal{L}_{\mathrm{cons}} = \frac{1}{|\mathbb{D}_U|} \sum_{\mathbf{x} \in \mathbb{D}_U} d\Big( p_{\mathrm{m}}\big(y \mid f_{\boldsymbol{\theta}}(\alpha(\mathbf{x})) - \Delta\mathfrak{R}\big), p_{\mathrm{m}}\big(y \mid f_{\boldsymbol{\theta}}(\mathcal{A}(\mathbf{x}))\big)\Big), \tag{2}$$

*where $\Delta\mathfrak{R}$ aims to pull $f_{\boldsymbol{\theta}}(\alpha(\mathbf{x}))$ closer to $f_{\boldsymbol{\theta}}(\mathcal{A}(\mathbf{x}))$. Ideally, $\Delta\mathfrak{R}$ would be adaptive, scaling with the discrepancy between $f_{\boldsymbol{\theta}}(\mathcal{A}(x))$ and $f_{\boldsymbol{\theta}}(\alpha(x))$, while avoiding make them too similar, which enables model to find a shortcut of $\mathcal{L}_{cons}$.*

To design it, we delve into the vision transformer, a representation encoder that has significantly advanced the performance of the GCD task. We found that the self-attention mechanism excels at capturing critical image patterns: as depicted in Fig. 3 left, deeper features (after the 8th layer) reveal semantic, high-level commonalities (*e.g.*, car shell) across all images; and the shallow features are more attuned to high-frequency, low-level details (*e.g.*, color and texture).

## 3.2 Inconsistent Patterns Spoil the Whole Barrel

Inconsistent patterns can disrupt learning, making it crucial to align stored and queried patterns effectively. To address this, we draw inspiration from the Hopfield Network [2] — an associative memory model known for its energy-based mechanism that naturally pulls similar patterns together

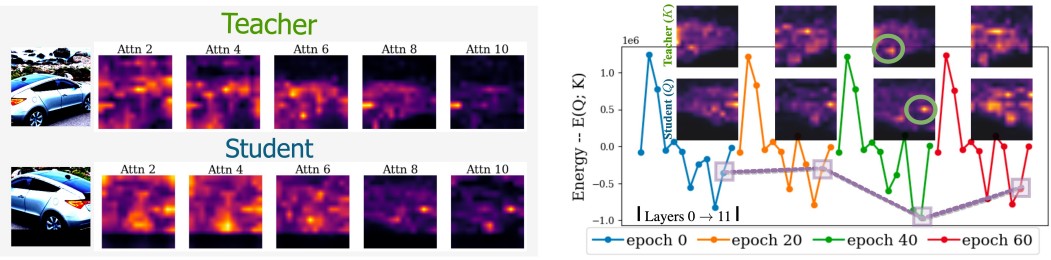

Figure 3: Left: Attention heatmaps for teacher and student across attention layers. Right: Energy trend over epochs, with lower energy indicating less discrepancy in pattern recognition between teacher and student.

(see *Appendix* A.1 for details). We follow Ramsauer et al. to define the energy function for a state pattern (query) $\boldsymbol{\xi} \in \mathbb{R}^d$, parameterized by $N$ stored (key) patterns $\mathbf{X} = [\mathbf{x}_1, \cdots, \mathbf{x}_N] \in \mathbb{R}^{d \times N}$:

$$E(\boldsymbol{\xi}; \mathbf{X}) = \frac{1}{2}\boldsymbol{\xi}^\top \boldsymbol{\xi} - \text{lse}(\mathbf{X}^\top \boldsymbol{\xi}, \beta) + c, \quad \text{with lse}(\mathbf{v}, \beta) := \beta^{-1} \log \left( \sum_{i=1}^{N} \exp(\mathbf{v}_i) \right). \quad (3)$$

Minimizing $E(\boldsymbol{\xi}; \mathbf{X})$ resembles retrieving stored pattern $\mathbf{x}_i$ that is most similar to the query $\boldsymbol{\xi}$, and *log-sum-exp* (lse) function is parameterized by $\beta > 0$ and $c$ is a preset constant. Particularly, the first term ensures the finiteness of the query, while the second term measures the alignment of the query with each stored pattern. The update rule for a state pattern $\boldsymbol{\xi}$ is equivalent to a gradient descent update of minimizing the energy $E$ with step size $\eta = 1$ [54], ensures that the query moves closer to the most similar stored pattern:

$$\boldsymbol{\xi} \leftarrow \boldsymbol{\xi} - \eta \nabla_{\boldsymbol{\xi}} E(\boldsymbol{\xi}; \mathbf{X}) = \boldsymbol{\xi} - \text{sm}(\beta \boldsymbol{\xi}^\top \mathbf{X}) \mathbf{X}^\top. \quad (4)$$

Moreover, the energy function is closely related to the Transformer's self-attention mechanism [67] (*Appendix* A.1.2). By extending the energy model from self-attention to cross-attention, we model the dynamics between student and teacher learning patterns. Taking the student representations $\mathbf{f}_s = f(\mathcal{A}(\mathbf{x}))$ as examples, we have $\mathbf{Q}_s = \mathbf{f}_s \mathbf{W}_Q$ and $\mathbf{K}_s = \mathbf{f}_s \mathbf{W}_K$. By applying Eq. 3 to key and query matrices, we set energy functions to track the teacher-student relationship:

$$E(\mathbf{Q}_s; \mathbf{K}_t) = \frac{\alpha}{2}\text{diag}(\mathbf{K_t K_t}^T) - \sum_{i=1}^{N} \text{lse}(\mathbf{Q}_s \mathbf{k}_{t,i}^T, \beta) + c, \quad (5a)$$

$$E(\mathbf{K}_t) = \text{lse}\left(\frac{1}{2}\text{diag}(\mathbf{K_t K_t}^T), 1\right) = \log \sum_{i=1}^{N} \exp\left(\frac{1}{2}\mathbf{k}_{t,i}\mathbf{k}_{t,i}^T\right) + c, \quad (5b)$$

where $E(\mathbf{Q}_s; \mathbf{K}_t)$ indicates the alignment in learning patterns of the student and teacher; $\mathbf{k}_{t,i}$ denotes the $i$-th row vector of $\mathbf{K}_t$ and $\alpha \geq 0$. Intuitively, $\text{lse}(\mathbf{Q}_s \mathbf{k}_{t,i}^T, \beta)$ captures the smooth maximum alignment between student queries $\mathbf{q}_{s,i}$ and teacher keys $\mathbf{k}_{t,i}$. Specifically, it nudges each teacher key $\mathbf{k}_{t,j}$ towards a semantic alignment with its most corresponding student query $\mathbf{q}_{s,j}$. The regularization term $\text{diag}(\mathbf{K_t K_t}^T)$ acts as a constraint on the energy levels of teacher's representation $\mathbf{k}_{t,i}$, guarding against any disproportionate increase during the maximization of $\text{lse}(\mathbf{Q}_s \mathbf{k}_{t,i}^T, \beta)$. This ensures that no individual teacher representation becomes too closely mirrored in the student's representation, maintaining a diverse learning trajectory.

**Insight 3.2.** *When applying closed-world consistency regularization to the GCD task, it becomes difficult to gradually reduce the energy $E(\mathbf{Q}_s; \mathbf{K}_t)$ as training progresses (Fig. 3 right). The sustained high energy demonstrated a flaw in the previous methods: teachers and students focused on identifying patterns that were inconsistent, leading to divergent learning paths. Specifically, when teachers and students focus on similar patterns (e.g., taillights), energy is reduced, indicating better prediction consistency and effective learning. In contrast, when their attention is distracted, the energy rises, leading to severe inconsistencies in predictions and making the optimization of $\mathcal{L}_{cons}$ more difficult.*

# 4 FlipClass: Teacher-Student Attention Alignment

## 4.1 Teacher Attention Update Rule

Based on Insights 3.1 and 3.2, our objective is to minimize the energy function $E(\mathbf{Q}_s; \mathbf{K}_t)$ between teacher and student representations, thereby easing the optimization of $\mathcal{L}_{\text{cons}}$.

**Theorem 4.1.** *The minimization can be formulated as obtaining a maximum a posteriori probability (MAP) estimate of teacher keys $\mathbf{K}_t$ given a set of observed student queries $\mathbf{Q}_s$:*

$$p(\mathbf{K}_t|\mathbf{Q}_s) = \frac{p(\mathbf{Q}_s|\mathbf{K}_t)p(\mathbf{K}_t)}{p(\mathbf{Q}_s)}, \tag{6}$$

*where $p(\mathbf{Q}_s|\mathbf{K}_t)$ and $p(\mathbf{K}_t)$ are modeled by energy functions Eq. 5a and 5b, respectively. We approximate the posterior inference by the gradient of the log posterior, estimated as:*

$$\begin{aligned}
\nabla_{\mathbf{K}_t} \log p(\mathbf{K}_t|\mathbf{Q}_s) &= -\left(\nabla_{\mathbf{K}_t} E(\mathbf{Q}_s; \mathbf{K}_t) + \nabla_{\mathbf{K}_t} E(\mathbf{K}_t)\right) \\
&= sm\left(\beta \mathbf{Q_s}\mathbf{K_t}^T\right)\mathbf{Q}_s - \left(\alpha\mathbf{I} + \mathcal{D}\left(sm\left(\frac{1}{2}diag(\mathbf{K_t}\mathbf{K_t}^T)\right)\right)\right)\mathbf{K}_t,
\end{aligned} \tag{7}$$

*where $sm(\mathbf{v}) := \exp\left(\mathbf{v} - lse(\mathbf{v}, 1)\right)$ and $\mathcal{D}(\cdot)$ is a vector-to-diagonal-matrix operator. Incorporating Eq. 4, the update rule of teacher keys $\mathbf{K}_t$ is derived as follows:*

$$\mathbf{K}_t^{update} = \mathbf{K}_t + \gamma_{update}\left[\left(sm\left(\beta\mathbf{K}\mathbf{Q}^T\right)\mathbf{Q}\mathbf{W}_K^T\right) - \gamma_{\text{reg}}\left(\alpha\mathbf{I} + \mathcal{D}\left(sm\left(\frac{1}{2}\operatorname{diag}\left(\mathbf{K}\mathbf{K}^T\right)\right)\right)\right)\mathbf{K}\mathbf{W}_K^T\right], \tag{8}$$

*where $\alpha$, $\gamma_{update}$ and $\gamma_{reg}$ are hyper-parameters.*

For a proof, refer to *Appendix* A.2. The teacher-attention update rule in Theorem 4.1 minimizes an implicit energy function determined by student queries and teacher keys. It serves as using the student queries to search for the most similar teacher patterns in the stored set. As illustrated in Fig. 4, the update rule adjusts the teacher's attention in the direction of student attention, facilitating the retrieval of related patterns and improving semantic alignment. This design establishes a bidirectional information flow: the teacher not only imparts advanced knowledge to the student, but also adjusts guidance based on the student's learning effects, achieving a more cohesive learning dynamic.

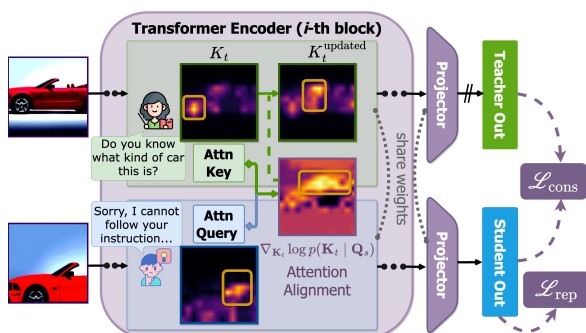

Figure 4: **Framework of *FlipClass*** demonstrating teacher-student interaction, where teacher's and student's attention is aligned by teacher's updating (Eq. 8). Then $\mathcal{L}_{\text{rep}}$ and $\mathcal{L}_{\text{cons}}$ are combined for optimization.

## 4.2 Representation Learning and Parametric Classification

Contrastive learning plus consistency regularization under the parametric paradigm has been demonstrated effective in GCD task [75]. Formally, given two views (random augmentations $\mathbf{x}_i$ and $\mathbf{x}'_i$) of the same image in a mini-batch $\mathbb{B}$, the supervised and self-supervised contrastive loss is written as:

$$\mathcal{L}_{\text{rep}}^s = \frac{1}{|\mathbb{B}'|}\sum_{i \in \mathbb{B}'}\frac{1}{|\mathbb{N}_i|}\sum_{q \in \mathbb{N}_i} -\log\frac{\exp(\mathbf{z}_i^T z'_q/\tau_c)}{\sum_{i' \neq i}\exp(\mathbf{z}_i^T z'_{i'}/\tau_c)},$$

$$\mathcal{L}_{\text{rep}}^u = \frac{1}{|\mathbb{B}|}\sum_{i \in \mathbb{B}} -\log\frac{\exp(\mathbf{z}_i^T \mathbf{z}'_i/\tau_u)}{\sum_{i' \neq i}\exp(\mathbf{z}_i^T z'_{i'}/\tau_u)},$$

where the feature $\mathbf{z}_i = f(\mathbf{x}_i)$ and is $\ell_2$-normalised, and $\tau_u$, $\tau_c$ are temperature values. For $\mathcal{L}_{\text{rep}}^s$, $\mathbb{N}_i$ indexes all other images in the same batch that hold the same label as $\mathbf{x}_i$. The representation learning loss is balanced with $\lambda$: $L_{\text{rep}} = (1-\lambda)L_{\text{rep}}^u + \lambda L_{\text{rep}}^s$, where $\mathbb{B}'$ corresponds to the labeled subset of $\mathbb{B}$.

Table 1: Evaluation on the Semantic Shift Benchmark (SSB). Bold values represent the best results, while underlined values represent the second-best results.

| Methods | Backbone | CUB | | | Stanford Cars | | | Aircraft | | | Avg. |
|---------|----------|-----|-----|-----|-----|-----|-----|-----|-----|-----|------|
| | | All | Old | New | All | Old | New | All | Old | New | |
| GCD [2022] | DINO | 51.3 | 56.6 | 48.7 | 39.0 | 57.6 | 29.9 | 45.0 | 41.1 | 46.9 | 45.1 |
| XCon [2022] | DINO | 52.1 | 54.3 | 51.0 | 40.5 | 58.8 | 31.7 | 47.7 | 44.4 | 49.4 | 46.8 |
| CiPR [2023] | DINO | 57.1 | 58.7 | 55.6 | 47.0 | 61.5 | 40.1 | - | - | - | - |
| PCAL [2023] | DINO | 62.9 | 64.4 | 62.1 | 50.2 | 70.1 | 40.6 | 52.2 | 52.2 | 52.3 | 55.1 |
| SimGCD [2023] | DINO | 60.3 | 65.6 | 57.7 | 53.8 | 71.9 | 45.0 | 54.2 | 59.1 | 51.8 | 56.1 |
| AdaptGCD [2024] | DINO | 66.6 | 66.5 | 66.7 | 48.4 | 57.7 | 39.3 | 53.7 | 51.1 | 56.0 | 56.2 |
| AMEND [2024] | DINO | 64.9 | 75.6 | 59.6 | 56.4 | 73.3 | 48.2 | 52.8 | 61.8 | 48.3 | 58.0 |
| GCA [2024] | DINO | 68.8 | 73.4 | 66.6 | 54.4 | 72.1 | 45.8 | 52.0 | 57.1 | 49.5 | 58.4 |
| TIDA [2024] | DINO | - | - | - | 54.7 | 72.3 | 46.2 | 54.6 | 61.3 | 52.1 | - |
| $\mu$GCD [2024] | DINO | 65.7 | 68.0 | 64.6 | 56.5 | 68.1 | 50.9 | 53.8 | 55.4 | 53.0 | 58.7 |
| CMS [2024] | DINO | 68.2 | 76.5 | 64.0 | 56.9 | 76.1 | 47.6 | 56.0 | 63.4 | 52.3 | 60.4 |
| InfoSieve [2024] | DINO | 69.4 | **77.9** | 65.1 | 55.7 | 74.8 | 46.4 | 56.3 | 63.7 | 52.5 | 60.5 |
| SPTNet [2024] | DINO | 65.8 | 68.8 | 65.1 | 59.0 | 79.2 | 49.3 | **59.3** | 61.8 | **58.1** | 61.4 |
| FlipClass (Ours) | DINO | **71.3** | 71.3 | **71.3** | **63.1** | **81.7** | 53.8 | **59.3** | 66.9 | 55.4 | **64.6** |
| Improvement | DINO | +5.5 | +2.5 | +6.2 | +4.1 | +2.5 | +4.5 | +0.0 | +5.1 | -2.7 | +3.2 |
| GCD [2022] | DINOv2 | 71.9 | 71.2 | 72.3 | 65.7 | 67.8 | 64.7 | 55.4 | 47.9 | 59.2 | 64.3 |
| SimGCD [2023] | DINOv2 | 71.5 | 78.1 | 68.3 | 71.5 | 81.9 | 64.6 | 49.9 | 60.9 | 60.0 | 63.0 |
| $\mu$GCD [2024] | DINOv2 | 74.0 | 75.9 | 73.1 | 76.1 | **91.0** | 68.9 | 66.3 | 68.7 | 65.1 | 72.1 |
| FlipClass (Ours) | DINOv2 | **79.3** | **80.7** | **78.5** | **78.0** | 88.0 | **73.2** | **71.1** | **75.1** | **69.1** | **76.1** |
| Improvement | DINOv2 | +5.3 | +4.8 | +5.4 | +1.9 | -3.0 | +4.3 | +4.8 | +6.4 | +4.0 | +4.0 |

The consistency regularization objectives (Eq. 1) are then simply cross-entropy loss $\ell(q', p) = -\sum_k q'(k) \log p(k)$ between the predictions and pseudo-labels or ground-truth labels:

$$\mathcal{L}_{\text{cons}} = \begin{cases} \frac{1}{|\mathbb{B}|} \sum_{i \in \mathbb{B}} \ell(q_i', p_i) - \varepsilon H(\bar{p}) & \text{for unlabeled,} \\ \frac{1}{|\mathbb{B}'|} \sum_{i \in \mathbb{B}'} \ell(y_i, p_i) & \text{for labeled.} \end{cases}$$

The one-hot labels $y_i$ correspond to $\mathbf{x}_i$, and the soft pseudo-label $q_i'$ is produced by the teacher instance $\alpha(\mathbf{x})_i$. Moreover, a mean-entropy regularizer [6], $H(\bar{p}) = -\sum_k \bar{p}(k) \log \bar{p}(k)$, is included to encourage diverse predictions. The combined classification loss, $\mathcal{L}_{\text{cons}}$, balances unsupervised and supervised terms with a parameter $\lambda$. And the overall training objective is $\mathcal{L}_{\text{rep}} + \mathcal{L}_{\text{cons}}$.

# 5 Experiments

## 5.1 Experimental Settings

**Datasets.** We evaluate the effectiveness of *FlipClass* on three generic image recognition datasets (*i.e.*, CIFAR-10/100 [34] and ImageNet-100 [20]), three fine-grained datasets [68] (*i.e.*, CUB [71], Stanford Cars [33], and FGVC-Aircraft [46]) contained in Semantic Shift Benchmark (SSB) [68], and the challenging datasets Herbarium-19 [65], ImageNet-1k [20]. For each dataset, we first subsample $|\mathbb{C}_l|$ seen (labeled) classes from all classes. Following GCD [69], we subsample 80% samples in CIFAR-100 and 50% samples in all other datasets from the seen classes to construct $\mathbb{D}_l$, while the remaining images are treated as $\mathbb{D}_u$ (refer to Table 9).

**Evaluation Protocols.** The performance was evaluated by measuring accuracy between the model's cluster assignments and ground-truth labels on the test set, with three aspects: all instances (All), instances from old categories (Old), and instances from new categories (New). The number of categories in the unlabeled dataset ($|\mathbb{C}_u|$) is often unknown. Following previous studies [64, 85], we set $K$ (cluster number) equal to $|\mathbb{C}_u|$, as approximate cluster estimation is usually feasible in the real world. The estimation of the number of categories in unlabeled datasets can be found in the *Appendix* C.4. Further implementation details can be found in *Appendix* D.1.

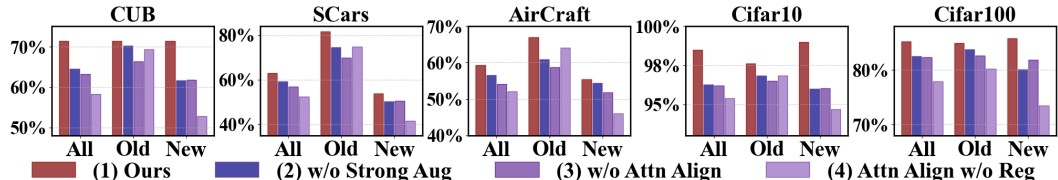

Figure 5: Ablation study results for FlipClass, indicate the critical role of strong augmentations, attention alignment, and regularization in model performance across multiple datasets.

Table 2: Evaluation on three generic image recognition datasets.

| Methods | Backbone | CIFAR10 | | | CIFAR100 | | | ImageNet-100 | | | Avg. |
|---|---|---|---|---|---|---|---|---|---|---|---|
| | | All | Old | New | All | Old | New | All | Old | New | |
| GCD [2022] | DINO | 91.5 | **97.9** | 88.2 | 73.0 | 76.2 | 65.5 | 74.1 | 89.8 | 66.3 | 79.5 |
| AdaptGCD [2024] | DINO | 93.2 | 94.6 | 92.8 | 71.3 | 75.7 | 66.8 | 83.3 | 90.2 | 76.5 | 82.6 |
| InfoSieve [2024] | DINO | 94.8 | 97.2 | 93.7 | 76.9 | 78.4 | 73.9 | 80.5 | 92.8 | 74.4 | 84.1 |
| CiPR [2023] | DINO | 97.7 | 97.5 | 97.7 | 81.5 | 82.4 | 79.7 | 80.5 | 84.9 | 78.3 | 86.6 |
| SimGCD [2023] | DINO | 97.1 | 95.1 | 98.1 | 80.1 | 81.2 | 77.8 | 83.0 | 93.1 | 77.9 | 86.7 |
| GCA [2024] | DINO | 95.5 | 95.9 | 95.2 | 82.4 | 85.6 | 75.9 | 82.8 | 94.1 | 77.1 | 86.9 |
| TIDA [2024] | DINO | 98.2 | **97.9** | 98.5 | 82.3 | 83.8 | 80.7 | - | - | - | - |
| CMS [2024] | DINO | - | - | - | 82.3 | **85.7** | 75.5 | 84.7 | **95.6** | 79.2 | - |
| AMEND [2024] | DINO | 96.8 | 94.6 | 97.8 | 81.0 | 79.9 | 83.8 | 83.2 | 92.9 | 78.3 | 87.0 |
| SPTNet [2024] | DINO | 97.3 | 95.0 | 98.6 | 81.3 | 84.3 | 75.6 | 85.4 | 93.2 | 81.4 | 88.0 |
| FlipClass (Ours) | DINO | **98.5** | 97.6 | **99.0** | **85.2** | 84.9 | **85.8** | **86.7** | 94.3 | **82.9** | **90.1** |
| Improvement | DINO | +1.2 | +2.6 | +0.4 | +3.9 | +0.6 | +10.2 | +1.3 | +1.1 | +1.5 | +2.1 |
| ∗ GCD [2022] | DINOv2 | 95.2 | 97.8 | 93.9 | 77.3 | 82.8 | 66.1 | 81.3 | 94.3 | 74.8 | 84.6 |
| ∗ AMEND [2024] | DINOv2 | 97.7 | 96.6 | 98.3 | 83.5 | 83.0 | 84.5 | 87.3 | 95.1 | 83.4 | 89.5 |
| FlipClass (Ours) | DINOv2 | **99.0** | **98.2** | **99.4** | **91.7** | **90.4** | **94.2** | **91.0** | **96.3** | **88.3** | **93.9** |
| Improvement | DINOv2 | +1.3 | +1.6 | +1.1 | +8.2 | +7.4 | +9.7 | +3.7 | +1.2 | +4.9 | +4.3 |

## 5.2 Experimental Results

We compare SOTA methods with ours in GCD using features from both DINO [15] and DINOv2 [51]. Our approach shows significant performance improvement, particularly in the recognition of 'New' classes across both the SSB fine-grained benchmark (Tab. 1) and generic image recognition datasets (Tab. 2), consistently surpassing existing SOTA methods. Moreover, in fine-grained image classification (Tab. 1), recognizing subtle differences between closely related categories is crucial, which is in contrast to coarse-grained datasets where the visual differences between classes are more obvious. In fine-grained settings, the risk of the model generating incorrect pseudo labels is higher, which makes consistency regularization counterproductive (Sec. 2.2). However, the results across these datasets demonstrate our model's capability to effectively adapt consistency regularization strategies from closed-world settings to more complex open-world scenarios. Moreover, the balanced accuracy between new and old classes on the CUB dataset (Tab. 1), also observed with methods like PCAL, CiPR, and AdaptGCD, can be attributed to the dataset's small size (6,000 images) and large class split (200). This limited data reduces the likelihood of overfitting to old classes, promoting more uniform performance across both categories. Additional results on Herbarium 19 and ImageNet-1k are detailed in the *Appendix* C.2.

## 5.3 Analysis and Discussion

**Ablation Studies.** Our ablation study, shown in Fig. 5, underscores the significance of our design choices. First, we replace the student's augmentations with the teacher's, *i.e.*, using only weak augmentations, the performance on 'New' classes significantly declines (2nd set of bars). This underscores the importance of **strong augmentations** for the student, which are essential to bolster generalizability. Then we validate the importance of attention alignment in Eq. 8 (3rd set of bars), we see performance drop across both 'Old' and 'New' classes, affirming that our attention alignment strategy is crucial for maintaining a consistent learning pace between the teacher and student, leading to sustained performance gains. Finally, the 4th set of bars verifies the role of **regularization** during

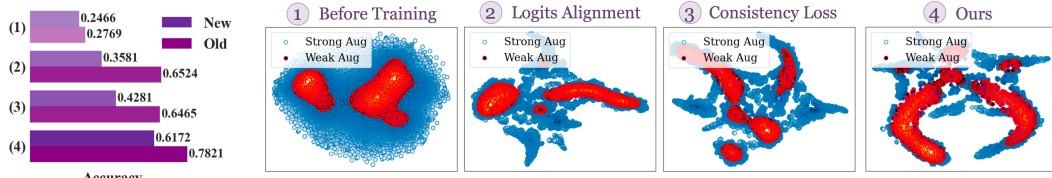

Figure 6: Accuracy and representation alignment with different strategies: (1) initial state, (2) distribution alignment, (3) FixMatch, and (4) our teacher-attention update. Performance on 'New' and 'Old' classes are shown, alongside alignment of teacher (red) and student (blue) representation.

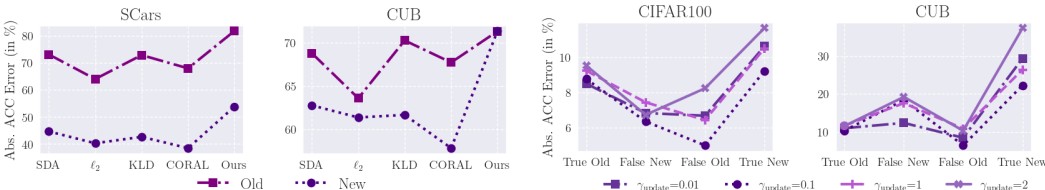

(a) Comparison of attention alignment methods, showcasing the effectiveness of our teacher-attention update strategy over alternatives in improving classification accuracy for 'Old' (solid) and 'New' (dotted) classes.

(b) Categorize errors for FlipClass on CIFAR100 and CUB, showing the reduction in prediction bias for 'False Old' and 'False New' classes with different update rates ($\gamma_{update}$), highlighting model robustness.

Figure 7: Attention alignment methods comparison and categorize errors with different update rates.

the attention update, which integrates the prior energy of the teacher, preventing any single student pattern from overly influencing the teacher's attention.

**Enhanced Consistency through Attention Alignment.** To validate our assumption on modeling $\mathfrak{R}$ in Eq. 2, we showcase how distribution-based strategies (distribution alignment [11], FixMatch [63]) and our representation-based method, attention alignment, achieve consistency on new classes from CUB dataset (Fig. 6). Our method stands out by significantly reducing the discrepancy between the representations of teacher (weakly-augmented) and student (strongly-augmented) data. This representation-based alignment leads to a more consistent learning process and is evidenced by the superior accuracies we achieve for both 'Old' and 'New' classes.

**Design Choice of Attention Alignment.** We experimented with various techniques to model $\mathfrak{R}$ in Eq. 2, including scheduled data augmentation (SDA), increasing similarity between $\mathbf{Q}_s$ and $\mathbf{K}_t$ via $\ell_2$ norm, Kullback-Leibler divergence (KLD) or CORrelation ALignment (CORAL) loss (see details in *Appendix* D.1). As shown in Fig. 7a, our teacher-attention update strategy outperforms these alternatives on both CUB and SCars datasets.

**FlipClass mitigates prediction bias.** We verify the effectiveness and robustness of *FlipClass*, by diagnosing the model's classification errors under four different ($\gamma_{update}$) as defined in Eq. 8. As depicted in Fig. 7b, both "False New" and "False Old" errors are consistently mitigated—where 'Old' class samples are mistakenly labeled as 'New' and vice-versa. Moreover, as illustrated in Fig.8b bottom, *FlipClass* outperforms leading methods [55] by moving closer to the true class distribution, yielding higher and less biased accuracies across all classes.

**Does the improved energy dynamic make for performance gains?** Fig. 8a top shows that aligning attention in these deeper (9-10) layers yields the highest performance on SCars dataset. And Fig. 8a bottom displays a greater reduction of energy $E(\mathbf{Q}_s; \mathbf{K}_t)$ in deeper layers. Moreover, this trend highlights that attention alignment constantly maintains lower energy levels than without, indicating improved alignment of student patterns with teacher updating.

**How does FlipClass change the representations?** Fig. 8b showcases representation enhancements with *FlipClass* against the leading method, InfoSieve [55], using the same t-SNE and PCA components to ensure consistent projection space and scale. FlipClass forms clusters with higher compactness and purity, demonstrating enhanced feature discrimination and less inter-class confusion. The zoom-in comparison on CUB dataset is provided in Fig. 9, which showcases that FlipClass achieves more distinct and well-separated clusters. Further, we assess prediction bias and class-

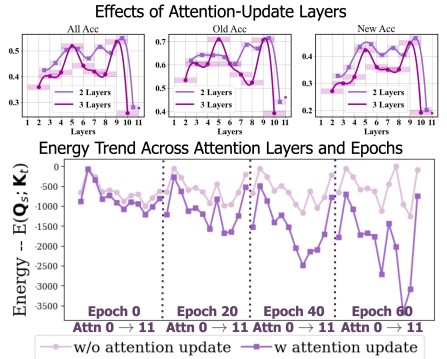
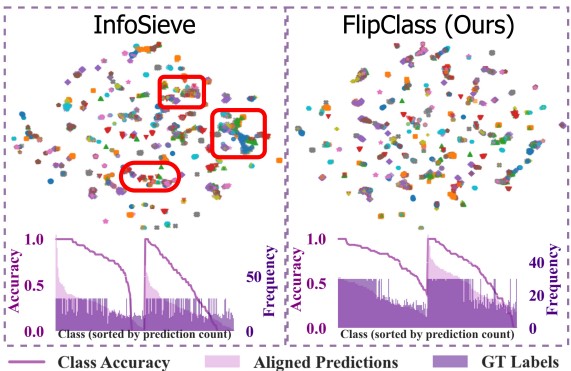

(a) Top: performance regarding attention-update layers. Bottom: Energy tracking across attention layers and epochs. (Both on SCars dataset)

(b) Comparison of representations and classwise accuracy between InfoSieve [55] and FlipClass on CUB.

Figure 8: Attention alignment improves energy dynamic and brings performance gains.

specific accuracies. Unlike InfoSieve's skewed predictions, *FlipClass* aligns better with true class distributions, and significantly improves over the tail classes in CUB dataset (More experiments in *Appendix* C.3).

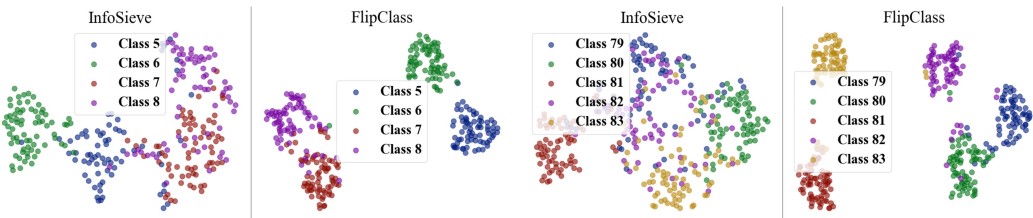

Figure 9: Zoom-in comparison of InfoSieve and FlipClass on the CUB dataset using t-SNE and PCA. FlipClass shows improved cluster separation and compactness.

## 6 Conclusion

This paper introduces *FlipClass*, a dynamic teacher-student attention alignment strategy for improving learning synchronization, providing a new view on applying closed-world models to open-world task of GCD. By aligning the attention of teacher and student, *FlipClass* bridges the learning gap between them, resulting in performance improvement on both old and new classes. Extensive experiments and analysis demonstrate that *FlipClass* outperforms existing state-of-the-art methods across diverse datasets by a large margin.

## 7 Acknowledgement

This work was supported by National Science and Technology Major Project (2022ZD0117102), National Natural Science Foundation of China (62293551, 62377038,62177038,62277042). Project of China Knowledge Centre for Engineering Science and Technology, Project of Chinese academy of engineering "The Online and Offline Mixed Educational Service System for 'The Belt and Road' Training in MOOC China". "LENOVO-XJTU" Intelligent Industry Joint Laboratory Project.

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

# Appendix

## Table of Contents

# A   Theory Assumptions and Proofs

## A.1   Preliminaries: Hopfield Network Energy Function

### A.1.1   Global Convergence of Hopfield Network Energy Function

We first provide the formulation of Hopfield network energy function, and present its convergence, which build the fundamental of our one-step Teacher-Attention Update Strategy. Ramsauer et al. proposed a new energy function that is a modification of the energy of modern Hopfield networks [19], and a new update rule which can be proven to converge to stationary points of the energy.

Given $N$ stored (key) patterns $\mathbf{x}_i \in \mathbb{R}^d$ represented by the matrix $\mathbf{X} = (\mathbf{x}_1, \ldots, \mathbf{x}_N)$ with the state (query) pattern $\boldsymbol{\xi} \in \mathbb{R}^d$, the energy function $E$ of the modern Hopfield networks can be expressed:

$$E = \exp(\text{lse}(1, \mathbf{X}^T \boldsymbol{\xi})),$$

where $\text{lse}(\beta, x) = \beta^{-1} \log \left( \sum_{i=1}^N \exp(\beta \mathbf{x}_i) \right)$ is the log-sum-exp function (lse) for $0 < \beta$. Ramsauer et al. then proposed to take the logarithm of the negative energy of modern Hopfield networks and add a quadratic term of the current state to ensure that the norm of the state vector $\boldsymbol{\xi}$ remains finite and the energy is bounded, reads:

$$E = -\text{lse}(\beta, \mathbf{X}^T \boldsymbol{\xi}) + \frac{1}{2} \boldsymbol{\xi}^T \boldsymbol{\xi} + \beta^{-1} \log N + \frac{1}{2} M^2. \tag{9}$$

Using $p = \text{softmax}(\beta \mathbf{X}^T \boldsymbol{\xi})$, the update rule is defined as:

$$\boldsymbol{\xi}^{\text{new}} = f(\boldsymbol{\xi}) = \mathbf{X}p = \mathbf{X}\text{softmax}(\beta \mathbf{X}^T \boldsymbol{\xi}), \tag{10}$$

which is the Concave-Convex Procedure (CCCP) for minimizing the energy $E$ and can be proven as converging globally.

**Theorem A.1** (Global Convergence (Zangwill): Energy). *The update rule Eq. 10 converges globally: For $\boldsymbol{\xi}^{t+1} = f(\boldsymbol{\xi}^t)$, the energy $E(\boldsymbol{\xi}^t) \to E(\boldsymbol{\xi}^*)$ for $t \to \infty$ and a fixed point $\boldsymbol{\xi}^*$.*

*Proof.* The Concave-Convex Procedure (CCCP) [81] minimizes a function that is the sum of a concave function and a convex function. And since *lse* is a convex, $-\text{lse}$ a concave function. Therefore, the energy function $E(\boldsymbol{\xi})$ is the sum of the convex function $E_1(\boldsymbol{\xi}) = \frac{1}{2} \boldsymbol{\xi}^T \boldsymbol{\xi} + C_1$ and the concave function $E_2(\boldsymbol{\xi}) = -\text{lse}$:

$$E(\boldsymbol{\xi}) = E_1(\boldsymbol{\xi}) + E_2(\boldsymbol{\xi}), \tag{11}$$

$$E_1(\boldsymbol{\xi}) = \frac{1}{2} \boldsymbol{\xi}^T \boldsymbol{\xi} + \beta^{-1} \ln N + \frac{1}{2} M^2 = \frac{1}{2} \boldsymbol{\xi}^T \boldsymbol{\xi} + C_1,$$

$$E_2(\boldsymbol{\xi}) = -\text{lse}(\beta, \mathbf{X}^T \boldsymbol{\xi}),$$

where $C_1$ does not depend on $\boldsymbol{\xi}$.

The Concave-Convex Procedure (CCCP) applied to $E$ is

$$\nabla_{\boldsymbol{\xi}} E_1(\boldsymbol{\xi}^{t+1}) = -\nabla_{\boldsymbol{\xi}} E_2(\boldsymbol{\xi}^t), \tag{12}$$

which results in the update rule:

$$\boldsymbol{\xi}^{t+1} = \mathbf{X}p^t = X\text{softmax}(\beta \mathbf{X}^T \boldsymbol{\xi}^t) \tag{13}$$

where $p^t = \text{softmax}(\beta \mathbf{X}^T \boldsymbol{\xi}^t)$. This is the update rule in Eq. 10.   □

### A.1.2   Hopfield Update Rule is Attention of The Transformer

The Hopfield network update rule is the attention mechanism used in transformer. Assume $N$ stored (key) patterns $\mathbf{x}_i$ and $S$ state (query) patterns $\mathbf{r}_i$ with dimension $d_k$, we can have $\mathbf{X} = (\mathbf{x}_1, \ldots, \mathbf{x}_N)^T$ and $\mathbf{R} = (\mathbf{r}_1, \ldots, \mathbf{r}_S)^T$ combine the $\mathbf{x}_i$ and $\mathbf{r}_i$ as row vectors. Define the key as $\mathbf{k}_i = \mathbf{W}_K^T \mathbf{x}_i$, $\mathbf{q}_i = \mathbf{W}_Q^T \mathbf{r}_i$, and multiply the result of the update rule (Eq. 10) with $\mathbf{W}_V$. By defining the matrices

$\mathbf{K} = (\mathbf{X}\mathbf{W}_K)^T$, $\mathbf{Q} = (\mathbf{R}\mathbf{W}_Q)^T$, and $\mathbf{V} = \mathbf{X}\mathbf{W}_K\mathbf{W}_V = \mathbf{K}^T\mathbf{W}_V$, where $\mathbf{W}_K \in \mathbb{R}^{d_x \times d_k}$, $\mathbf{W}_Q \in \mathbb{R}^{d_r \times d_k}$, $\mathbf{W}_V \in \mathbb{R}^{d_k \times d_o}$. If $\beta = 1/\sqrt{d_k}$ and softmax $\in \mathbb{R}^N$ is changed to a row vector, there is:

$$\text{softmax}\left(\frac{1}{\sqrt{d_k}}\mathbf{Q}\mathbf{K}^T\right)\mathbf{V} = \text{softmax}\left(\beta\mathbf{R}\mathbf{W}_Q\mathbf{W}_K^T\mathbf{X}^T\right)\mathbf{X}\mathbf{W}_K\mathbf{W}_V, \tag{14}$$

where the left part is the transformer attention, while the right part is the update rule Eq. 4 multiplied by $\mathbf{W}_V$.

## A.2 Derivation of The Teacher Attention Update Rule

We first provide several lemmas for the derivation of the teacher-attention update rule (Eq. 8).

**Lemma A.2.** *For a given column vector $\mathbf{x} \in \mathbb{R}^N$, we have:*

$$\frac{\partial \log \sum \exp(\beta\mathbf{x})}{\partial \mathbf{x}} = \text{softmax}(\beta\mathbf{x}) \tag{15}$$

*Proof.* Consider $S = \sum_{i=1}^{N} \exp(\beta\mathbf{x}_i)$ and $f(\mathbf{x}) = \log S$, $\frac{\partial f(\mathbf{x})}{\partial \mathbf{x}}$ can be computed as:

$$\frac{\partial f(\mathbf{x})}{\partial \mathbf{x}_j} = \frac{\partial}{\partial \mathbf{x}_j} \log S = \frac{1}{S} \cdot \frac{\partial S}{\partial \mathbf{x}_j}$$

The partial derivative $\frac{\partial S}{\partial \mathbf{x}_j}$ is:

$$\frac{\partial S}{\partial \mathbf{x}_j} = \frac{\partial}{\partial \mathbf{x}_j} \sum_{i=1}^{N} \exp(\beta\mathbf{x}_i) = \sum_{i=1}^{N} \frac{\partial}{\partial \mathbf{x}_j} \exp(\beta\mathbf{x}_i) = \sum_{i=1}^{N} \beta \exp(\beta\mathbf{x}_i)\delta_{ij} = \beta \exp(\beta\mathbf{x}_j)$$

where $\delta_{ij}$ is the Kronecker delta.

Substitute $\frac{\partial S}{\partial \mathbf{x}_j}$ back into the expression for $\frac{\partial f(\mathbf{x})}{\partial \mathbf{x}_j}$:

$$\frac{\partial f(\mathbf{x})}{\partial \mathbf{x}_j} = \frac{1}{S} \cdot \beta \exp(\beta\mathbf{x}_j) = \beta \cdot \frac{\exp(\beta\mathbf{x}_j)}{S}$$

Recognize that $\frac{\exp(\beta\mathbf{x}_j)}{S}$ is the $j$-th component of the softmax function applied to $\beta\mathbf{x}$:

$$\text{softmax}(\beta\mathbf{x})_j = \frac{\exp(\beta\mathbf{x}_j)}{\sum_{i=1}^{N} \exp(\beta\mathbf{x}_i)} = \frac{\exp(\beta\mathbf{x}_j)}{S}$$

Therefore, we have:

$$\frac{\partial \log \sum_{i=1}^{N} \exp(\beta\mathbf{x}_i)}{\partial \mathbf{x}_j} = \beta \cdot \text{softmax}(\beta\mathbf{x})_j$$

Putting it back into vector notation:

$$\frac{\partial \log \sum_{i=1}^{N} \exp(\beta\mathbf{x}_i)}{\partial \mathbf{x}} = \beta \cdot \text{softmax}(\beta\mathbf{x})$$

Since $\frac{\partial \log \sum \exp(\beta\mathbf{x})}{\partial \mathbf{x}} = \beta^{-1}\frac{\partial \log \sum_{i=1}^{N} \exp(\beta\mathbf{x}_i)}{\partial \mathbf{x}}$, we can have:

$$\frac{\partial \log \sum \exp(\beta\mathbf{x}_i)}{\partial \mathbf{x}} = \text{softmax}(\beta\mathbf{x})$$

This confirms the lemma. $\square$

**Lemma A.3.** *Let $\mathbf{k}_i$ denote the $i$-th row vector of $\mathbf{K} \in \mathbb{R}^{N \times d}$. Then, we have:*

$$\frac{\partial \mathbf{k}_i\mathbf{k}_i^T}{\partial \mathbf{K}} = 2\mathbf{e}_i^N(\mathbf{e}_i^N)^T\mathbf{K}, \tag{16}$$

*where $\mathbf{e}_i^N$ represents an $N$-dimensional column vector where only the $i$-th entry is 1, with all other entries set to zero.*

*Proof.* First, note that the expression $\mathbf{k}_i\mathbf{k}_i^T$ can be equivalently rewritten as $\mathbf{e}_i^N\mathbf{k}_i\mathbf{k}_i^T(\mathbf{e}_i^N)^T$.

To find the derivative of $\mathbf{k}_i\mathbf{k}_i^T$ with respect to $\mathbf{K}$, we need to consider the individual elements of $\mathbf{K}$. Let $\mathbf{K} = [\mathbf{k}_1^T; \mathbf{k}_2^T; \cdots; \mathbf{k}_N^T]$. Therefore, $\mathbf{k}_i^T$ is the $i$-th row of $\mathbf{K}$, and we denote this as $\mathbf{k}_i^T = \mathbf{K}_{i,:}$.

The differential of $\mathbf{k}_i\mathbf{k}_i^T$ is given by:

$$d(\mathbf{k}_i\mathbf{k}_i^T) = d(\mathbf{K}_{i,:}^T\mathbf{K}_{i,:}) = d(\mathbf{K}_{i,:}^T)\mathbf{K}_{i,:} + \mathbf{K}_{i,:}^T d(\mathbf{K}_{i,:}).$$

Since $\mathbf{K}_{i,:}^T d(\mathbf{K}_{i,:}) = \mathbf{e}_i^N(\mathbf{e}_i^N)^T d\mathbf{K}\mathbf{K}_{i,:} = (\mathbf{e}_i^N(\mathbf{e}_i^N)^T d\mathbf{K})\mathbf{K}$ and similarly for the transpose term, we get:

$$d(\mathbf{k}_i\mathbf{k}_i^T) = \mathbf{e}_i^N(\mathbf{e}_i^N)^T d\mathbf{K}\mathbf{K}_{i,:} + \mathbf{K}_{i,:}^T(\mathbf{e}_i^N(\mathbf{e}_i^N)^T d\mathbf{K}).$$

Thus, we can summarize the derivative as:

$$\frac{\partial\mathbf{k}_i\mathbf{k}_i^T}{\partial\mathbf{K}} = 2\mathbf{e}_i^N(\mathbf{e}_i^N)^T\mathbf{K},$$

which matches Eq. 16 □

**Lemma A.4.**
$$\nabla_{\mathbf{K}}\mathrm{lse}(\mathbf{Q}\mathbf{k}_i^T, \beta) = \mathrm{softmax}(\beta\mathbf{Q}\mathbf{k}_i^T)\cdot\mathbf{Q} \tag{17}$$

*Proof.* Define $\mathbf{z} = \mathbf{Q}\mathbf{k}_i^T$, we can rewrite $\mathrm{lse}(\mathbf{Q}\mathbf{k}_i^T, \beta)$ as:

$$\mathrm{lse}(\mathbf{z}, \beta) = \beta^{-1}\log\left(\sum_{j=1}^{N}\exp(\beta z_j)\right)$$

Using Lemma A.3, we have:
$$\nabla_{\mathbf{z}}\mathrm{lse}(\mathbf{z}, \beta) = \mathrm{softmax}(\beta\mathbf{z}) \tag{18}$$

Substitute $\mathbf{z} = \mathbf{Q}\mathbf{k}_i^T$ back to the expression, we have:

$$\nabla_{\mathbf{K}}\mathrm{lse}(\mathbf{Q}\mathbf{k}_i^T, \beta) = \nabla_{\mathbf{z}}\mathrm{lse}(\mathbf{z}, \beta)\cdot\frac{\partial\mathbf{z}}{\partial\mathbf{K}}$$

With $\frac{\partial z_j}{\partial\mathbf{k}_i} = \mathbf{Q}_{j,:}$ and Eq. 18, we have:

$$\nabla_{\mathbf{K}}\mathrm{lse}(\mathbf{Q}\mathbf{k}_i^T, \beta) = \mathrm{softmax}(\beta\mathbf{Q}\mathbf{k}_i^T)\cdot\mathbf{Q}$$

Thus, the lemma is proved. □

**Lemma A.5.**
$$\frac{\partial\mathrm{diag}(\mathbf{K}\mathbf{K}^T)}{\partial\mathbf{K}} = 2\mathbf{K} \tag{19}$$

*where diag($\mathbf{A}$) denotes the trace of $\mathbf{A}$.*

*Proof.* Let us construct a column vector $\mathbf{x}$ whose $i$-th element is given by $\mathbf{x}_i := \mathbf{k}_i\mathbf{k}_i^T/2$. Then, using Lemmas A.2 and A.3 and the chain rule, we have:

$$\begin{aligned}
\frac{\partial\mathrm{diag}(\mathbf{K}\mathbf{K}^T)}{\partial\mathbf{K}} &= \frac{\partial\log\sum_{i=1}^{N}\exp\left(\frac{1}{2}\mathbf{k}_i\mathbf{k}_i^T\right)}{\partial\mathbf{K}} \\
&= \sum_i\frac{\partial\mathbf{x}_i}{\partial\mathbf{K}}\frac{\partial\mathrm{lse}(\mathbf{x}, 1)}{\partial\mathbf{x}_i} \\
&= \sum_i\mathbf{e}_i^N(\mathbf{e}_i^N)^T\mathbf{K}[\mathrm{softmax}(x)]_i \qquad \text{Because of Lemma A.2 and A.3} \\
&= \sum_i\mathbf{e}_i^N\mathbf{k}_i[\mathrm{softmax}(\mathbf{x})]_i = 2\mathbf{K}
\end{aligned}$$

This proves the lemma. □

**Derivation of Eq. 7.** Now we proof that we can approximate the posterior inference of $p(\mathbf{K}_t \mid \mathbf{Q}_s)$ by the gradient of the log posterior, estimated as:

$$\nabla_{\mathbf{K}_t} \log p(\mathbf{K}_t | \mathbf{Q}_s) = -\left(\nabla_{\mathbf{K}_t} E(\mathbf{Q}_s; \mathbf{K}_t) + \nabla_{\mathbf{K}_t} E(\mathbf{K}_t)\right)$$

$$= \mathrm{sm}\left(\beta \mathbf{Q_s} \mathbf{K_t}^T\right) \mathbf{Q}_s - \left(\alpha \mathbf{I} + \mathcal{D}\left(\mathrm{sm}\left(\frac{1}{2}\mathrm{diag}(\mathbf{K_t}\mathbf{K_t}^T)\right)\right)\right) \mathbf{K}_t,$$

where $\mathrm{sm}(\mathbf{v}) = \mathrm{softmax}(\mathbf{v}) := \exp(\mathbf{v} - \mathrm{lse}(\mathbf{v}, 1))$ and $\mathcal{D}(\cdot)$ is a vector-to-diagonal-matrix operator. Moreover, recall Eq. 5a and 5b, the energy $E(\mathbf{Q}_s; \mathbf{K}_t)$ and $E(\mathbf{K}_t)$ are denoted as:

$$E(\mathbf{Q}_s; \mathbf{K}_t) = \frac{\alpha}{2}\mathrm{diag}(\mathbf{K_t}\mathbf{K_t}^T) - \sum_{i=1}^{N} \mathrm{lse}(\mathbf{Q}_s \mathbf{k}_{t,i}^T, \beta) + c,$$

$$E(\mathbf{K}_t) = \mathrm{lse}\left(\frac{1}{2}\mathrm{diag}(\mathbf{K_t}\mathbf{K_t}^T), 1\right) = \log \sum_{i=1}^{N} \exp\left(\frac{1}{2}\mathbf{k}_{t,i}\mathbf{k}_{t,i}^T\right) + c,$$

*Proof.* Using Lemmas A.4 and A.5, we have:

$$\nabla_{\mathbf{K}_t} E(\mathbf{Q}_s; \mathbf{K}_t) = \alpha \mathbf{K}_t - \mathrm{sm}(\beta \mathbf{Q}_s \mathbf{K}_t^T) \cdot \mathbf{Q}$$

Since $\mathbf{K}_t$ can be expressed as $\mathrm{lse}(\frac{1}{2}\mathbf{k}_i \mathbf{k}_i^T)$, incorporating Lemma A.5, we have:

$$\nabla_{\mathbf{K}_t} E(\mathbf{K}_t) = \mathcal{D}\left(\mathrm{sm}\left(\frac{1}{2}\mathrm{diag}(\mathbf{K_t}\mathbf{K_t}^T)\right)\right) \mathbf{K}_t$$

Put them together, we have:

$$\nabla_{\mathbf{K}_t} \log p(\mathbf{K}_t | \mathbf{Q}_s) = \mathrm{sm}\left(\beta \mathbf{Q_s} \mathbf{K_t}^T\right) \mathbf{Q}_s - \left(\alpha \mathbf{I} + \mathcal{D}\left(\mathrm{sm}\left(\frac{1}{2}\mathrm{diag}(\mathbf{K_t}\mathbf{K_t}^T)\right)\right)\right) \mathbf{K}_t,$$

This matches Eq. 7. □

## B    Extended Experimental Analysis of Attention Alignment

In this section, we begin by examining the representation discrepancy between old and new classes, highlighting alignment issues (B.1). Enhanced consistency loss optimization is then detailed, showing improvements in learning stability (B.2). We discuss how attention alignment bridges the prior gap and benefits synchronized learning, enhancing overall performance (B.3). The focus then shifts to attention specialization in deep network layers (B.4), and the performance impact of layer selection for attention alignment on different dataset (B.5). Finally, we represent the negligible impact on computational cost of the Attention Alignment strategy (B.6), thereby proving its practical viability.

### B.1    Representation Discrepancy of Old and New Classes

In GCD, directly applying consistency regularization leads to challenges, especially for new classes. Due to the lack of prior knowledge, the teacher struggles to guide the student, resulting in representation discrepancies between the teacher (weakly-augmented) and student (strongly-augmented). This is evident in Fig. 10 (left), where new classes show poor alignment between teacher and student representations compared to known classes.

This discrepancy causes unsynchronized learning, as shown in Fig. 10 (right). The two main phenomena observed are the *learning gap* and *learning regression*. The *learning gap* indicates that the student struggles to reach the teacher's level of understanding, particularly for new classes, leading to stagnation. *Learning regression* affects the teacher, hampering improvement for new classes and causing regression in known classes due to the alignment efforts with the student.

### B.2    Enhanced Consistency Loss Optimization

Building on the previous discussion on the representation discrepancy of old and new classes, we address the challenges in optimizing consistency loss (Sec. 2.2) that contribute to the learning gap. To

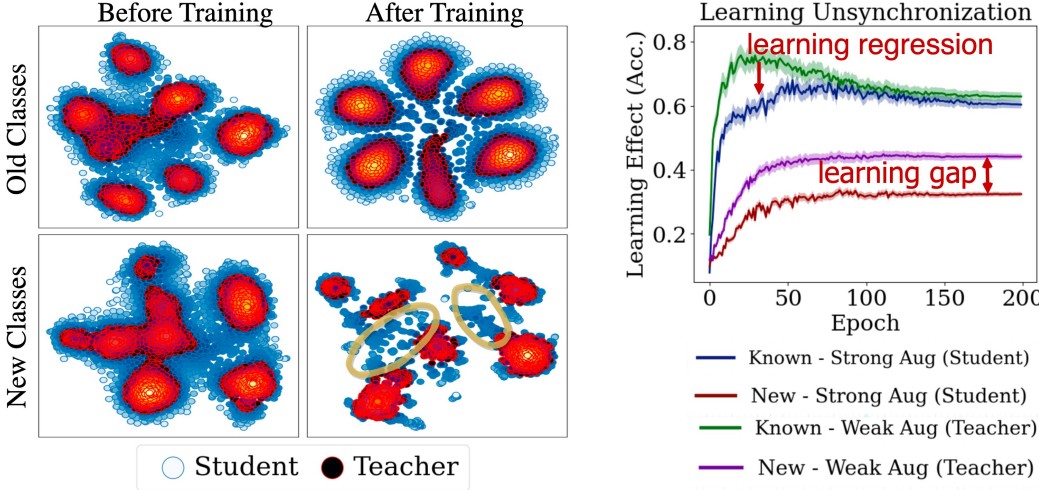

Figure 10: Left: Comparison of representation discrepancy with respect to old and new classes before and after training, showing the misalignment of student (blue) and teacher (red). Right: Learning unsynchronization between teacher and student with trends of *learning regression* and *learning gap* for old and new classes.

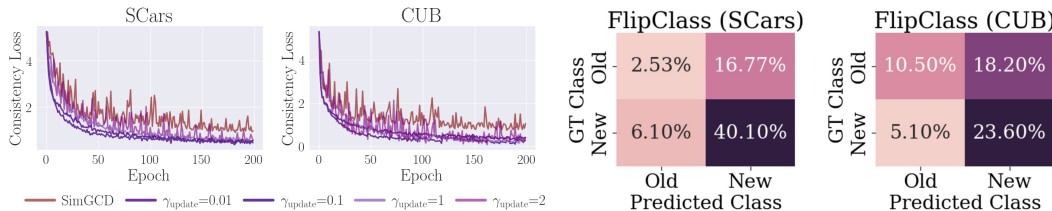

(a) **Consistency loss optimization on SCars and CUB**, comparing FlipClass with various update rates ($\gamma_{update}$) to the SimGCD baseline, demonstrating more rapid and stable convergence.

(b) **Categorize errors of *FlipClass***. Compared to those of SimGCD (Fig. 2 right), the reduced errors on 'False New' represent that FlipClass mitigates the overfitting of old classes brought by the prior gap.

Figure 11: Attention alignment bridges the prior gap with better-converged consistency loss and leads to less biased predictions.

verify how *FlipClass* improves this process, we track the consistency loss $\mathcal{L}_{cons}$ on new classes. As shown in Fig. 11a, FlipClass with various update rates ($\gamma_{update}$) demonstrates faster and more stable convergence compared to the SimGCD baseline. Specifically, the experiments on SCars and CUB datasets reveal that FlipClass streamlines the optimization process of consistency loss, leading to more rapid and stable convergence.

## B.3    Attention Alignment Bridges Prior Gap and Benefits Synchronized Learning

The enhanced consistency loss optimization further aids in achieving consistency between the student and the teacher, reflecting in two main aspects. Firstly, it **mitigates the effects of prior gap**, as shown in Fig. 11b. Compared to the categorize errors of SimGCD (Fig. 2), *FlipClass* reduces the 'False New' error (where the model incorrectly predicts new classes as old classes). This indicates that *FlipClass* can mitigate overfitting on old classes due to the lack of prior knowledge about new classes. Secondly, as shown in Fig. 12, compared to the traditional teacher-student model used in generalized category discovery (e.g., SimGCD), which suffers from a learning gap, *FlipClass* successfully **bridges the learning gap**. It achieves better teacher-student learning effects, ensuring more synchronized and stable learning outcomes.

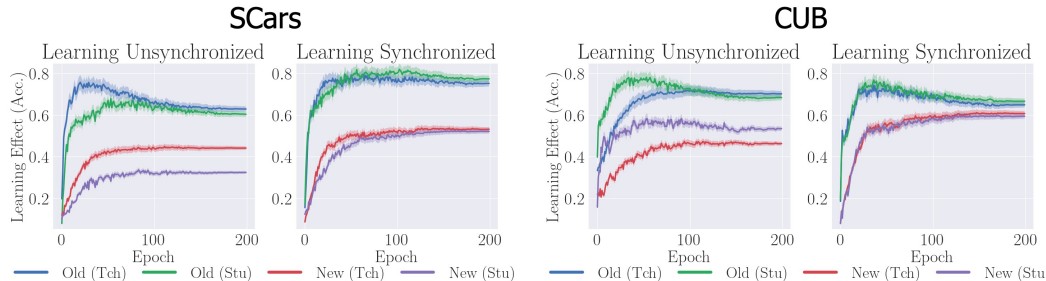

Figure 12: **Learning curves for SCars and CUB datasets**. *FlipClass* achieves better synchronized and stable learning effects compared to the traditional teacher-student model.

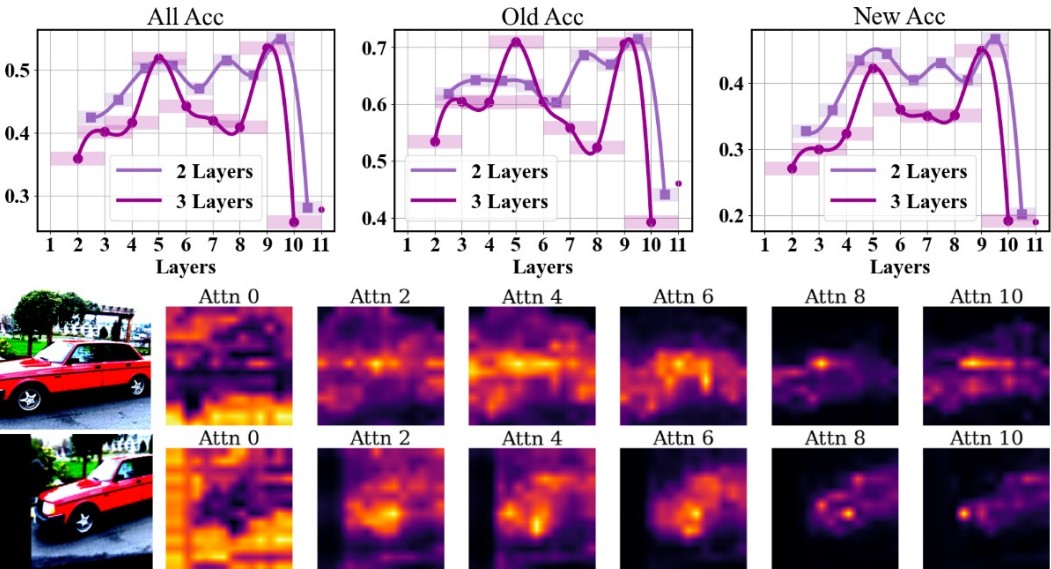

Figure 13: **Attention heatmap accuracy per layer on SCars dataset**, with deeper layers focused on local semantic features and earlier layers on general features, indicating better transfer learning for old and new classes with attention alignment in deeper network layers.

## B.4 Attention Specialization in Deep Network Layers

Moreover, on the SCars dataset, the model shows a greater energy reduction in deeper layers, suggesting a focus on local semantics over general ones. This is illustrated in Fig. 13, where deeper layers emphasize local semantic parts, and earlier ones are associated with general semantics (e.g., texture, color). This local semantic concentration enhances transferability across 'Old' and 'New' classes, with attention alignment in deeper layers (9-10) yielding the highest performance. We observe that different heads attend to disjoint regions of the image, focusing on important parts. After training with our method, attention heads become more specialized to semantic parts, displaying more concentrated and local attention. Our model learns to specialize attention heads (shown as columns) to different semantically meaningful parts, improving transferability between labeled and unlabeled categories.

## B.5 Attention Alignment in Layer Selection

Furthermore, we provide the performance of layer selection for attention alignment on CUB and Cifar-10. In this context, "2 layers" means performing attention alignment in the two layers depicted in the Fig. 14. As shown, Cifar-10 (coarse-grained) tends to achieve higher performance when alignment is performed in the middle layers (4-5), while CUB (fine-grained) achieves higher performance with alignment in deeper layers, which aligns with the trend of SCars (Fig. 13). The difference stems from

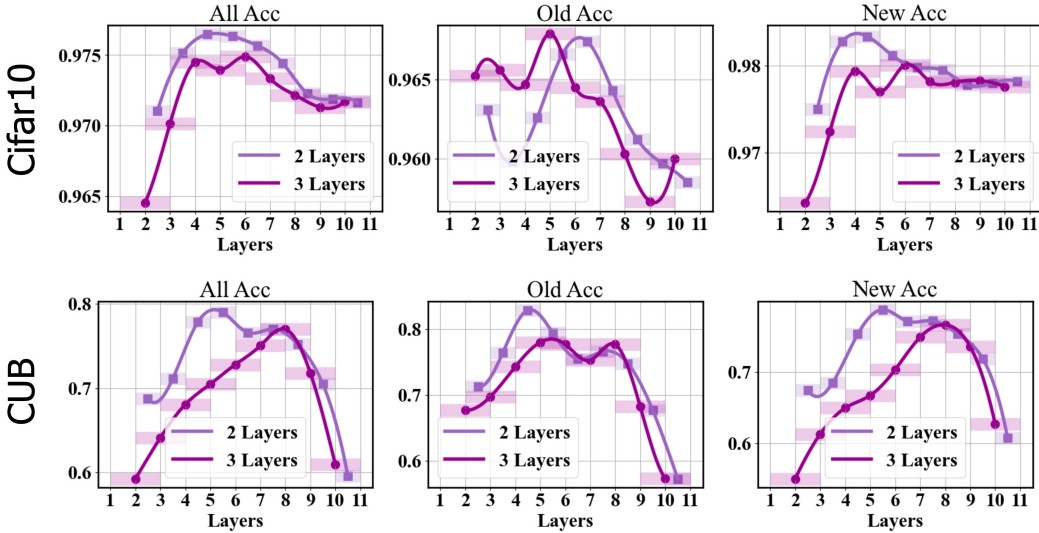

Figure 14: Layer selection performance for attention alignment on Cifar-10 and CUB datasets. Higher performance is observed in middle layers for Cifar-10 and deeper layers for CUB, indicating the need to tailor attention alignment to the nature of the data.

the dataset nature. CIFAR-10 benefits from middle-layer alignment, capturing general features like shapes and textures, which suffice for its simpler categories. Conversely, CUB requires deeper layer alignment for detailed features needed to distinguish similar bird species. Deeper layers provide the refined features critical for complex tasks.

## B.6 Time Efficiency of Attention Alignment

A potential concern is whether attention alignment increases the time cost. Since we perform the alignment in a one-loop manner, the additional time overhead is negligible. This is demonstrated in Table 3, showing that the training and inference times for FlipClass are comparable to those of SimGCD.

Table 3: Comparison of training and inference times for SImGCD on ImageNet-100 and AirCraft datasets, with 200 epochs

| Method | ImageNet-100 | | AirCraft | |
|---|---|---|---|---|
| | Training time (Min) | Infer. time (Sec) | Training time (Min) | Infer. time (Sec) |
| SimGCD [75] | 1639 | 591 | 224 | 17 |
| **FlipClass (Ours)** | 1648 | 591 | 229 | 17 |

We further analyze the potential extra overhead of our attention update strategy compared to a conventional self-attention block:

- **Overall Training Cost**: For a Vision Transformer (ViT) with $L$ layers, the training cost includes (1) *self-attention cost*, $O(L \cdot N \cdot d_k^2)$; (2) *feed-forward network (FFN) cost*, $O(L \cdot N \cdot d_k \cdot d_m)$. Here, $N$ is the sequence length, $d_k$ is the feature dimension, and $d_m$ is the hidden dimension in the FFN.
- **Attention Update Cost**: Our update strategy applies only to 2-3 layers (see Appendix B.5), with each update having the same cost as one FFN operation per layer.

In sum, we add **2-3 extra FFN-like operations** during the forward pass. And there is no impact on the backward pass, as only the last block is updated without introducing extra parameters for optimization. Therefore, the additional overhead is minimal, with only a slight increase in training time, as shown in Table 4.

Table 4: Time cost (seconds) per forward pass on CUB and Stanford Cars.

| Setting | CUB | SCars |
|---|---|---|
| w/o Attention Updating | 65s | 93s |
| w/ Attention Updating (2 layers) | 71s | 101s |
| w/ Attention Updating (3 layers) | 76s | 109s |

Table 5: Complete results of *FlipClass* in have five independent runs with random seeds.

| Dataset | All | | Old | | New | |
|---|---|---|---|---|---|---|
| | Mean | Std | Mean | Std | Mean | Std |
| CIFAR10 | 98.5 | ± 0.1 | 97.6 | ± 0.2 | 99.0 | ± 0.4 |
| CIFAR100 | 85.2 | ± 0.3 | 84.9 | ± 0.5 | 85.8 | ± 1.2 |
| ImageNet-100 | 86.7 | ± 0.5 | 94.3 | ± 1.2 | 82.9 | ± 0.9 |
| CUB | 71.3 | ± 1.4 | 71.3 | ± 3.2 | 71.3 | ± 2.0 |
| Stanford Cars | 63.1 | ± 0.7 | 81.7 | ± 1.8 | 53.8 | ± 1.7 |
| FGVC-Aircraft | 59.3 | ± 1.98 | 66.9 | ± 3.66 | 55.4 | ± 4.20 |

In summary, the attention update operation introduces a small extra overhead equivalent to **2-3 additional FFN operations**, which remains insignificant relative to the overall training cost, especially given the performance improvements achieved by FlipClass.

## C    More Experimental Results

We detail our experimental results, including main outcomes with error bars for statistical significance (C.1). We evaluate performance on complex datasets (C.2), analyze clustering and per-class prediction distributions (C.3). Additionally, we examine robustness to varying numbers of old and new classes (C.4), and investigate how different proportions of old classes affect performance (C.5), demonstrating our method's stability and adaptability.

### C.1    Main Results with Error Bars

Table 5 reports error bars to provide a clear understanding of the statistical significance and variability of the main results in our experiments. Specifically, we include both the mean and standard deviation (std) values for the performance of our *FlipClass* across different datasets and class types (All, Old, New). The standard deviations are calculated from five independent runs with random seeds, offering insight into the consistency of our method's performance.

### C.2    Results on Complex Datasets

Here, we discuss the performance of various methods on challenging datasets, specifically Herbarium19 [65] and ImageNet-1K [35]. Herbarium19, a long-tailed dataset, poses significant challenges due to the varying frequencies of different categories, leading to unbalanced cluster sizes. Table 6 demonstrates the robustness of our proposed *FlipClass*, in handling such frequency imbalances and its ability to accurately distinguish categories even with few examples. Table 7 showcases the performance on ImageNet-1K, a large-scale generic classification dataset, in evaluating the model's capability in real-world applications of generalized category discovery. The results demonstrate the robustness of *FlipClass* for tasks involving both familiar and unfamiliar data in complex, real-world scenarios.

### C.3    Clustering and Per-class Prediction Distribution

**Clustering Analysis.** Fig. 15 presents a visual comparison of the clustering results obtained with *FlipClass* against the existing state-of-the-art method, InfoSieve [55], on Cifar-10 and Cifar-100 datasets. On Cifar-10, FlipClass forms clusters that exhibit higher compactness and purity, indicating

Table 6: Performance comparison of different methods on the Herbarium19 dataset.

| Methods | Pretraining | Herbarium19 | | |
| --- | --- | --- | --- | --- |
| | | **All** | **Known** | **Novel** |
| k-means [1967] | DINO | 13.0 | 12.2 | 13.4 |
| ORCA [2021] | DINO | 20.9 | 30.9 | 15.5 |
| RS+ [2020] | DINO | 27.9 | 55.8 | 12.8 |
| UNO+ [2021] | DINO | 28.3 | 53.7 | 12.8 |
| GCD [2022] | DINO | 35.4 | 51.0 | 27.0 |
| CMS [2024] | DINO | 36.4 | 54.9 | 26.4 |
| OpenCon [2022] | DINO | 39.3 | 58.9 | 28.6 |
| InfoSieve [2024] | DINO | 41.0 | 55.4 | 33.2 |
| MIB [2022] | DINO | 42.3 | 56.1 | 34.8 |
| PCAL [2023] | DINO | 37.0 | 52.0 | 28.9 |
| SimGCD [2023] | DINO | 44.0 | 58.0 | 36.4 |
| AMEND [2024] | DINO | 44.2 | 60.5 | 35.4 |
| $\mu$GCD [2024] | DINO | 45.8 | **61.9** | 37.2 |
| FlipClass (Ours) | DINO | **46.3** | 60.2 | **40.7** |

Table 7: Performance comparison of different methods on the ImageNet-1K dataset.

| Methods | Pretraining | ImageNet-1K | | |
| --- | --- | --- | --- | --- |
| | | **All** | **Known** | **Novel** |
| GCD [69] | DINO | 52.5 | 72.5 | 42.2 |
| SimGCD [2023] | DINO | 57.1 | 77.3 | 46.9 |
| FlipClass (Ours) | DINO | **59.2** | **78.9** | **49.5** |

enhanced feature discrimination and reduced interclass confusion. In contrast, on Cifar-100, although InfoSieve forms visually more compact clusters, these clusters show less purity, with a higher incidence of false class predictions.

**Prediction Distribution and Class-specific Accuracies.** Fig. 16 evaluates the prediction distribution and class-specific accuracies of *FlipClass* compared to InfoSieve [55]. *FlipClass* demonstrates a better fit to the true distribution, whereas InfoSieve shows skewed predictions. Moreover, *FlipClass* significantly outperforms InfoSieve in recognizing tail classes on the CUB and Stanford Cars datasets, improving accuracy and reducing prediction bias.

### C.4 Robustness to Number of Classes

**Varying Number of Classes during Clustering.** In the main experiments (Section 5), the class number, $K$, is assumed as a known prior following prior works [69, 64, 75, 70], however, this setting has been questioned as impractical [9, 74, 73]. In Fig. 17, we conduct experiments when this assumption is removed, evaluating results with different numbers of classes, where the ratio changes from 80% to 200% compared to the ground truth number of classes. During clustering (e.g., KMeans [45]), a predefined class number lower than the ground truth significantly limits the ability to discover new classes, causing the model to focus more on old classes. Conversely, increasing the class number results in less harm to the generic image recognition datasets (e.g., Cifar-100) and can even be beneficial for some fine-grained, long-tailed datasets (e.g., CUB). This phenomenon occurs because overestimating the number of classes allows the model to maintain higher flexibility and adaptability in recognizing new classes in these fine-grained, challenging datasets. For fine-grained, class-distribution biased datasets like CUB, overestimating the class number helps capture subtle differences between closely related categories, thereby improving class separation and reducing prediction bias. However, for generic datasets like Cifar-100, the visual differences between classes are more pronounced, and overestimating the class number can introduce unnecessary complexity, leading to overfitting and decreased performance.

**Estimation of Number of Classes.** Additionally, to further validate the robustness of our model, we trained FlipClass using an estimated number of classes in the dataset, where the number of classes

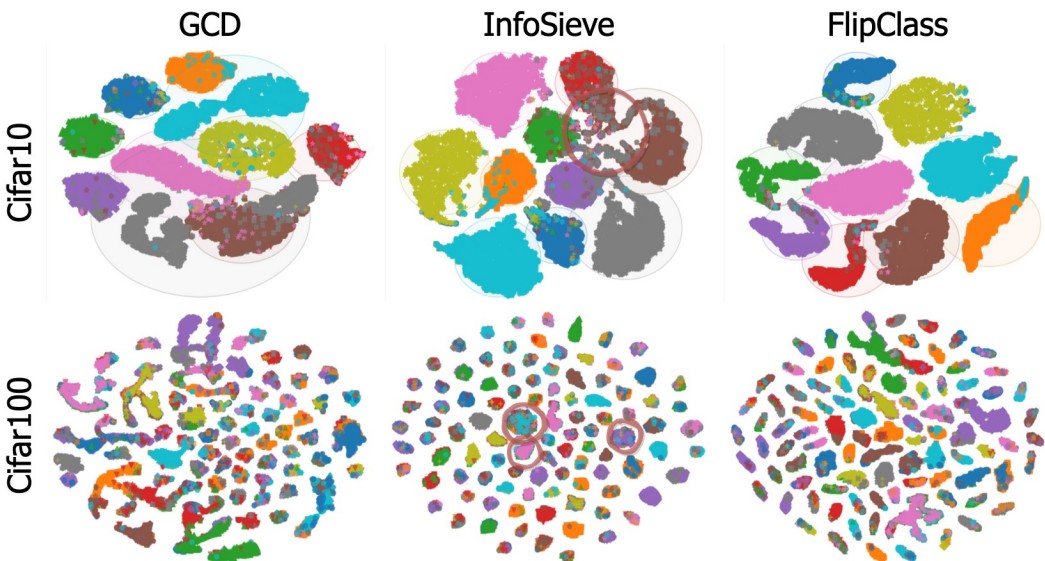

Figure 15: **Comparison of clustering results on Cifar-10 and Cifar-100** datasets using GCD, InfoSieve, and our *FlipClass*.

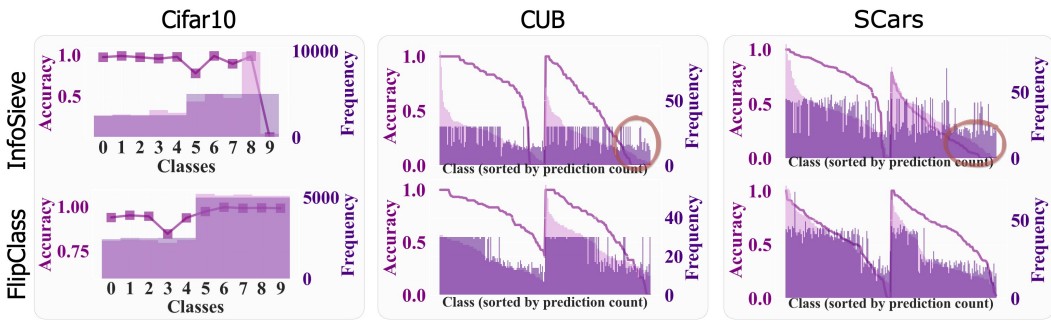

Figure 16: **Prediction distribution and class-specific accuracies** of InfoSieve and *FlipClass* on Cifar-10, CUB, and Stanford Cars datasets.

is predicted using the over-clustering method from GCD [69]. We obtained a similar predicted number of classes as SimGCD. As expected, our method performs worse on Cifar-100 when using an estimated number of classes due to the mismatch between the estimated and actual class distribution. Interestingly, the performance of SimGCD (traditional teacher-student model) improves on CUB with both new and old classes, while our FlipClass makes improvements on old classes but sees a decrease in new classes.

## C.5 Results with Varying Proportion of Old Classes

In the primary experiments, we fix the number of the old classes $|C_\ell|$ (details in *Appendix* D.3). Here, we experiment with our method by changing the class split setting. Specifically, on Cifar-100 ($|C_\ell| = 80$) and CUB ($|C_\ell| = 100$) datasets, we test with fewer old classes, as shown in Fig. 18. For Cifar-100 and CUB, as the number of old classes decreases, the accuracy for both old and new classes slightly declines but remains stable. This demonstrates *FlipClass*'s effectiveness in leveraging additional old class information and robustness in handling varying numbers of known classes.

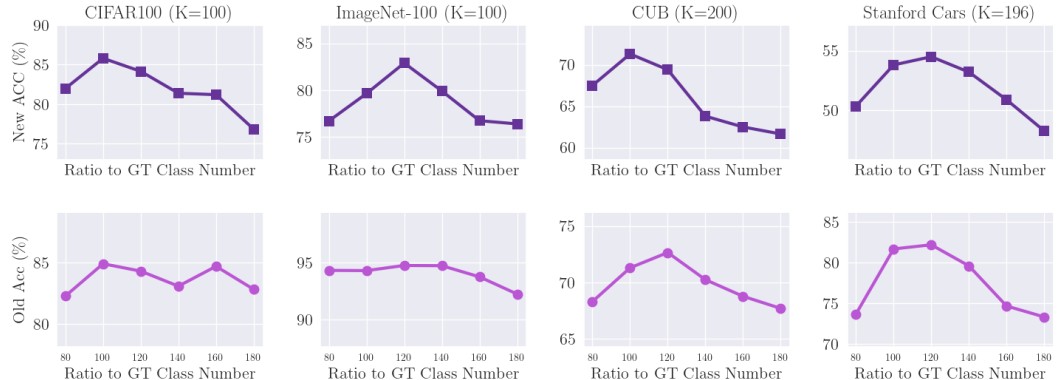

Figure 17: **Results with varying numbers of classes during clustering**, where the ratio changes from 80% to 200% of the ground truth number of classes.

Table 8: Performance of FlipClass and the baseline method SimGCD with an estimated number of categories on CUB and Cifar100. Bold values represent the best results.

| Method | $|C_{\text{all}}|/|C_\ell|$ | CUB | | | Cifar100 | | |
|---|---|---|---|---|---|---|---|
| | | All | Old | New | All | Old | New |
| SimGCD [75] | GT (200/100) | 60.3 | 65.6 | 57.7 | 80.1 | 81.2 | 77.8 |
| FlipClass (Ours) | GT (200/100) | **71.3** | 71.3 | **71.3** | **85.2** | **84.9** | **85.8** |
| SimGCD [75] | Est. (231/109) | 61.0 | 66.0 | 58.6 | 81.1 | 90.9 | 76.1 |
| FlipClass (Ours) | Est. (299/108) | 70.5 | **72.7** | 69.4 | 84.2 | 84.3 | 84.1 |

# D   Experimental Settings

## D.1   Implementation Details

We develop our FlipClass upon the SimGCD [75] baseline on the pre-trained ViT-B/16 DINO[2] [15]. Specifically, we take the final feature corresponding to the CLS token from the backbone as the image feature, which has a dimension of 768. For the feature extractor $\mathbb{F}$, we only fine-tune the last block. We set the balancing factor $\lambda$ to 0.35 and the temperature values $\tau_u$ and $\tau_c$ to 0.07 and 1.0, respectively, following SimGCD. For the temperature values $\tau_t$ and $\tau_s$ in the classification losses, we also set them to 0.07 and 0.1. For update rule (Eq. 8), we set $\alpha = 0$, $\beta = 1$, $\gamma_{\text{update}} = 0.1$ and $\gamma_{\text{reg}} = 0.5$. All experiments are conducted using a single NVIDIA A100 GPU with 200 epochs, which we find sufficient for the losses to plateau.

## D.2   Design of Data Augmentation

In this experimental setup, we design both weak and strong augmentations for the teacher and student networks. For weak augmentation, we use common techniques such as RandomHorizontalFlip and RandomCrop for all datasets, aiming to pass less perturbed versions of the input images to the teacher network. For the strong augmentation that is applied to the images fed to the student, we incorporate more aggressive transformations to expose the student to a wider range of variations. Specifically, we add RandomResizedCrop with a scale range of 0.3 to 1.0, which allows for more aggressive cropping and resizing. Additionally, we include Gaussian blurring to simulate different levels of image blurriness. For datasets that are used for generic recognition tasks, we further enhance the strong augmentation by including ColorJitter with probability 0.8 and RandomGrayscale with probability 0.2. Solarization inverts pixel values above a threshold, simulating the effect of solarizing an image, while Grayscale converts the image to black and white, reducing color information. These additional augmentations help expose the student network to even more diverse image variations, improving its robustness and generalization capabilities.

---

[2] https://huggingface.co/facebook/dino-vitb16

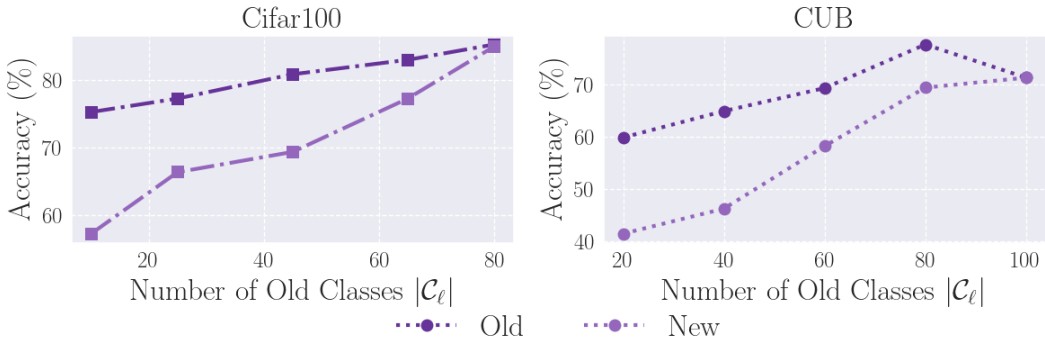

Figure 18: Results with varying the number of old classes $|\mathcal{C}_\ell|$.

## D.3 Datasets

Table 9: Statistics of the datasets used in our experiments.

|  | CIFAR10 | CIFAR100 | ImageNet100 | CUB200 | Stanford Cars | Aircraft | Herbarium 19 |
|---|---|---|---|---|---|---|---|
| $|\mathcal{Y}^l|$ | 5 | 80 | 50 | 100 | 98 | 50 | 341 |
| $|\mathcal{Y}^u|$ | 10 | 100 | 100 | 200 | 196 | 100 | 683 |
| $|\mathcal{D}^l|$ | 12.5k | 20k | 31.9k | 1.5k | 2.0k | 1.7k | 8.9k |
| $|\mathcal{D}^u|$ | 37.5k | 30k | 95.3k | 4.5k | 6.1k | 5.0k | 25.4k |

We follow the dataset settings from earlier works [75, 70] to subsample the training dataset. Specifically, 50% of known categories and all samples of unknown categories are used for training. For all datasets except Cifar-100, 50% of the categories are considered known during training, whereas for Cifar-100, 80% of the categories are known during training. Detailed statistics are displayed in Table 9. Below is a summary of the datasets and how their combination supports experiments in generalized category discovery:

*Cifar-10/100*[3] [34] are coarse-grained datasets consisting of general categories with low-resolution images and even class distribution.

*ImageNet-100/1K*[4] is a subset of 100/1K categories from the coarse-grained ImageNet[5] [35, 58] dataset. It includes a large scale of high-resolution real-world images with evenly distributed classes.

*CUB* (Caltech-UCSD Birds-200-2011)[6] [71] is widely used for fine-grained image recognition, containing different bird species distinguished by subtle details.

*Stanford Cars*[7] [33] is a fine-grained dataset of various car brands, providing multi-view objects for class detection and scene understanding, challenging real-world applications in distinguishing subtle appearance differences.

*FGVC-Aircraft* (Fine-Grained Visual Classification of Aircraft)[8] [46] is a fine-grained dataset organized in a three-level hierarchy. At the finer level, differences between models are subtle but visually measurable. Unlike animals, aircraft are rigid and less deformable, presenting variations in purpose, size, designation, structure, historical style, and branding.

---

[3]https://www.cs.toronto.edu/~kriz/cifar.html
[4]https://www.kaggle.com/c/imagenet-object-localization-challenge/overview/description
[5]https://www.image-net.org/download.php
[6]https://www.vision.caltech.edu/datasets/cub_200_2011/
[7]https://www.kaggle.com/datasets/jessicali9530/stanford-cars-dataset
[8]https://www.robots.ox.ac.uk/~vgg/data/fgvc-aircraft/

*Herbarium 19* (FGVC 2019 Herbarium Challenge)[9] [65] provides a curated dataset of over 46,000 herbarium specimens across 680 species, presenting a long-tailed distribution and challenges for species recognition.

### D.4   Other Alignment Strategies

In the ablation studies (Section 5.3), we apply different strategies such as distribution alignment [11], FixMatch [63] and CORAL. These strategies are employed to encourage the consistency between the teacher and student, therefore modeling $\mathfrak{R}$ in Eq. 2. We briefly provide the main idea of these alignment strategies below.

**Distribution alignment** is designed by maintaining a running average of the model's predictions on unlabeled data $\tilde{p}(y)$. Given the model's prediction $q = p_{\text{model}}(y \mid \mathbf{x}_u)$ on an unlabeled example $\mathbf{x}_u$, $q$ is scaled by the ratio $p(y)/\tilde{p}(y)$ and then renormalize the result to form a valid probability distribution: $\tilde{q} = \text{Normalize}(q \times p(y)/\tilde{p}(y))$.

**FixMatch** is a semi-supervised learning method that combines consistency regularization and pseudo-labeling. It works by first generating pseudo-labels for unlabeled images using the model's predictions on weakly enhanced versions of those images, retaining only high-confidence predictions as following:

$$\ell_u = \frac{1}{\mu B} \sum_{b=1}^{\mu B} \mathbb{1}\left(\max\left(q_b\right) \geq \tau\right) \mathrm{H}\left(\hat{q}_b, p_{\mathrm{m}}\left(y \mid \mathcal{A}\left(u_b\right)\right)\right),$$

where $\tau$ is a scalar hyperparameter denoting the threshold above which a pseudo-label should be retained. Then, the model is trained to predict these pseudo-labels using strongly enhanced versions of the same images. The loss function consists of two cross-entropy terms: a supervised loss for labeled data and an unsupervised loss for unlabeled data, where the unsupervised loss utilizes the pseudo-labels calculated from weakly enhanced images and the model's predictions on strongly enhanced images.

**CORAL** (CORelation ALignment) aligns the second-order statistics (covariances) of two spaces. Specifically, CORAL minimizes the difference in covariance matrices between the student and teacher representations ($\mathbf{S}$ and $\mathbf{T}$). The goal of CORAL is to find a transformation for $\mathbf{S}$ that minimizes the Frobenius norm of the difference between the covariance matrices of $\mathbf{S}$ and $\mathbf{T}$, denoting $\mathbf{C}_S$ and $\mathbf{C}_T$, respectively. CORAL minimizes the following objective:

$$\min_{S'} \|\mathbf{C}_{S'} - \mathbf{C}_T\|_F^2,$$

where $\mathbf{C}_{S'}$ is the covariance matrix of the transformed student representations $\mathbf{S}'$, and $\|.\|_F$ denotes the Frobenius norm.

## E   Related Works

### E.1   Consistency Regularization

In semi-supervised learning (SSL), the goal is to enhance model performance by leveraging unlabeled data, traditionally drawn from the same class spectrum as the labeled data [90]. A key strategy in SSL, consistency regularization [36], in recent years, centers on promoting model stability by ensuring that the teacher instance (weakly-augmented instance) and the student instance (strongly-augmented instance) yield coherent predictions[11, 63, 78, 87, 83]. Building on the $\Pi$-Model's teacher-student framework, several approaches have advanced its capabilities [66, 43, 77, 12, 53]. MeanTeacher [66] deploys an exponential moving average of the model parameters to stablize the teacher's output. NoisyStudent [77] employs a self-training strategy that incorporates noise into the student model's training, cycling the improved student back into the teacher role. Previous methods in SSL have largely concentrated on promoting the teacher's performance, often overlooking whether the student can keep pace, and neglecting the harmony of interaction. Our approach pivots to synchronizing the teacher's and student's attention, a shift that's especially pivotal in GCD, where consistency is challenged by the introduction of new classes. This strategy ensures a balanced teacher-student dynamic, crucial for effective consistency regularization in the open-world setting.

---

[9]`https://www.kaggle.com/c/herbarium-2019-fgvc6`

### E.2 Novel Category Discovery

Novel category discovery (NCD) is first formalized as cross-task transfer in [30], which aims to discover unseen categories from unlabeled data that have nonoverlapped classes with the labeled ones. Earlier works [31, 26, 86, 79, 38] mostly maintain two networks for learning from labeled and unlabeled data respectively. AutoNovel [27] introduces a three-stage framework. Specifically, the model is firstly trained with the whole dataset in a self-supervised manner and then fine-tuned only with the fully-supervised labeled set to capture the semantic knowledge for the final joint-learning stage. UNO [22] addresses the problem by jointly modeling the labeled and unlabeled sets to prevent the model from overfitting to labeled categories. Similarly, NCL [88] generates pairwise pseudo labels for unlabeled data and mixes samples in the feature space to construct hard negative pairs.

### E.3 Generalized Category Discovery

Generalized Category Discovery (GCD) extends NCD by categorizing unlabeled images from both seen and unseen categories [69], which tackles this issue by tuning the representation of the pre-trained ViT model with DINO ([15], [51]) with contrastive learning, followed by semi-supervised k-means clustering. ORCA [13] considers the problem from a semi-supervised learning perspective and introduces an adaptive margin loss for better intra-class separability for both seen and unseen classes. CiPR [28] introduces a method for more effective contrastive learning and a hierarchical clustering method for GCD without requiring the category number in the unlabeled data to be known a priori. SimGCD [75] proposes a parametric method with entropy regularization to improve performance. TIDA [74] discovers multi-granularity semantic concepts and then leverages them to enhance representation learning and improve the quality of pseudo labels. Moreover, $\mu$GCD [70] take a leap forward by extending the MeanTeacher paradigm to the GCD task. Instead of managing dual models as in $\mu$GCD, our approach achieves teacher-student consistency more effectively within a single-model structure, streamlining computational demands. Crucially, we found that the learning discrepancy between teachers and students in the open-world context is the reason why consistency is difficult to achieve, and solved this problem by synchronizing the attention of teachers and students.

## F  Limitations and Future Work

**Catastrophic Forgetting.** While our method achieves significant improvement on new classes, the performance on old classes, particularly on CUB, lacks compared to the state-of-the-art methods. We attribute this to a phenomenon akin to catastrophic forgetting, where the model forgets previously learned concepts. Addressing these issues is essential for enhancing the robustness and effectiveness of the proposed methods.

**Sub-optimal $K$ Estimation.** As shown in Appendix C.4, for fine-grained, class-distribution biased datasets like CUB, overestimating the class number helps capture subtle differences between closely related categories, thereby improving class separation and reducing prediction bias. However, for coarse-grained datasets like CIFAR-100, overestimating the class number can introduce unnecessary complexity, leading to overfitting and decreased performance. Some works have delved into this path and show promising performance [73, 74, 9], highlighting the potential of tailored $K$ estimation strategies to balance complexity and performance across different types of datasets.

**Data Augmentation to Enhance Teacher-Student Consistency**. Effective data augmentation techniques has been investigated a lot in semi-supervised learning [18, 82, 48, 76, 29], the techniques for generalized category discovery are still lacking, which affects the consistency between the teacher and student models. The strength of data augmentation for new classes needs careful control to avoid ineffective learning due to excessive noise or insufficient variability. Additionally, preventing data leakage during augmentation is critical, as pretrained diffusion models can compromise evaluation integrity by leaking training data. Addressing these issues is essential for enhancing the robustness and effectiveness of the proposed methods.

## G  Broader Impacts

Our study extends the capability of AI systems from the closed world to the open world, fostering AI systems capable of categorizing and organizing open-world data automatically. While Generalized

Category Discovery (GCD) has many real-world applications, it can be unreliable and must be applied with caution. Currently, supervised learning with extensive fine annotations is the mainstream solution for many computer vision tasks, but the cost and difficulty of obtaining these annotations can be prohibitive. Our work addresses this issue by advancing an open-set semi-supervised learning paradigm, significantly reducing the need for precise annotations and promoting the application of AI models in areas where annotations are difficult to obtain.

This work provides a new idea for open-set semi-supervised learning. Specifically, while conventional approaches apply closed-world semi-supervised learning techniques to generalized category discovery, they rarely consider the attention alignment gap between teacher and student models. We point out that bridging this gap can significantly improve learning efficiency and accuracy. We hope that this methodology can be generalized to more relevant label-efficient tasks, promoting broader applications of AI in scenarios with limited labeled data.

