# OpenReview forum: "Flipped Classroom: Aligning Teacher Attention with Student in Generalized Category Discovery"
_NeurIPS.cc/2024/Conference — NeurIPS 2024 oral_

### Official Review · Reviewer_Btrz · 2024-07-07

**Soundness:** 3
**Presentation:** 3
**Contribution:** 3
**Rating:** 6
**Confidence:** 3

**Summary:**

This paper targets the task of generalized class discovery (GCD), and argues that the existing teacher-student learning framework suffers from three challenges: 1) learning gap between old and new classes, 2) feature discrepancies between augmented images, and 3) attention inconsistency between teacher and student. These challenges, originating from inadequate supervision on new classes and the gap between weakly and strongly augmented data, largely hinder the performance of existing methods. In light of this, this paper proposes both empirical and theoretical analysis of the aforementioned issues, and introduces a novel method that can synchronize the learning progress of teacher and student models, which largely improves the GCD performance.

**Strengths:**

1. The paper is well organized and clearly written. The motivation is also strong, which is focused on largely-ignored perspectives in GCD literature.
1. The paper delivers useful empirical and theoretical insights, associated with an effective plug-and-play solution.
1. The experimental results showcase the superiority of the proposed method, which largely surpasses the existing SOTA methods.

**Weaknesses:**

1. Given the strong performance shown in Tables 1 and 2, I'm curious about what cost it takes to achieve that, since the modification is simply a new attention operation that offers the ability to sync the learning paces of teacher and student models. In this regard, I'd like to know the potential extra computation overhead (e.g., theoretical/empirical analysis) compared with the conventional self-attention block.

**Questions:**

Please see above.

**Limitations:**

The limitations are adequately discussed.

---

> ### Author Rebuttal · Authors · 2024-08-07
>
> Thanks for your thorough review. We appreciate your attention to detail regarding the potential extra computation overhead compared with conventional self-attention block.
>
> **Response to Q1**
> > ***Potential extra computation overhead (e.g., theoretical/empirical analysis) compared with the conventional self-attention block.***
>
> Here is a detailed comparison of the computation overhead:
>
> **1.Overall Training Computation Cost**:
> Considering $L$ layers, the computation costs for a Vision Transformer (ViT) model are as follows:
>
> 1. **Self-Attention Cost:** $O(B \cdot N^2 \cdot d)$
> 2. **Feed-Forward Network (FFN) Cost:** $O(B \cdot N \cdot d^2)$
>
> **2.Attention Update Computation Cost**:
>
> Our attention update strategy is applied to only 2-3 layers (the choice of layers are explained in L261-265 and Appendix B.5), and each updating cost is the same as the FFN cost per layer:
> $$ O(B \cdot N \cdot d^2) $$
>
> **3.Comparison of Overheads:**
>
> During the forward pass, we **add only 2-3 extra FFN-like operations**. \
> This updating does not affect the backward pass, as we only update the last block and no extra parameters are introduced to optimize.
>
> **4.Empirical Analysis:**
>
> The additional overhead is minimal, with only a slight increase in training time (table below), demonstrating that the computational cost is negligible compared to the performance improvements achieved by *FlipClass*.
>
> | Setting                         | CUB  | SCars |
> | ------------------------------- | ---- | ----- |
> | w/o Attention Updating          | 65s  | 93s   |
> | w Attention Updating (2 layers) | 71s  | 101s  |
> | w Attention Updating (3 layers) | 76s  | 109s  |
> *Time Cost (s) of One Forward Pass on CUB and Stanford Cars.*
>
> In summary, **the attention update operation introduces a small extra overhead equivalent to 2-3 extra FFN operations, which is insignificant compared to the overall training cost**.
>
> ---
> \
> **We have promptly integrated this analysis into our manuscript to enhance the clarity. Once again, we extend our sincere appreciation for your insightful feedback.**

---

> ### Author Response · Authors · 2024-08-13
>
> Dear Reviewer Btrz,
>
> Thank you again for reviewing our manuscript. We have tried our best to address your questions (see our rebuttal above), and revised our paper by following suggestions from all reviewers.
>
> Please kindly let us know if you have any follow-up questions or areas needing further clarification. Your insights are valuable to us, and we stand ready to provide any additional information that could be helpful.

---

### Official Review · Reviewer_P1mR · 2024-07-11

**Soundness:** 3
**Presentation:** 2
**Contribution:** 4
**Rating:** 7
**Confidence:** 4

**Summary:**

The paper introduces FlipClass, a novel method addressing the challenges of Generalized Category Discovery (GCD) in open-world scenarios. It identifies the misalignment of attention between teacher and student models as a key issue hindering effective learning, especially when new classes are introduced. To tackle this, FlipClass dynamically updates the teacher's attention based on feedback from the student, promoting synchronized learning and consistent pattern recognition. Extensive experiments demonstrate that FlipClass outperforms contemporary GCD methods.

**Strengths:**

1. This paper proposes an algorithm for Generalized Category Discovery. The authors underscore the significance of representation alignment between teacher (weak augmented) and student (strong augmented).

2. The idea about aligning representation is interesting.

3. The authors have provided the source code for reproduction.

**Weaknesses:**

1. The content regarding "the Hopfield Network" and the underlying motivation could be enhanced，and the current presentation is not clear and optimal.

2. The SOTA method SPTNet [1] should also be compared in the paper.

3. The differences between distribution alignment [2,3] and representation alignment in the paper need further discussion to make it easier for readers to understand your contributions.

[1] SPTNet: An efficient alternative framework for generalized category discovery with spatial prompt tuning. ICLR 2024.
[2] Open-World Semi-Supervised Learning. ICLR 2022.
[3] Robust Semi-Supervised Learning when Not All Classes have Labels. NeurIPS 2022.

**Questions:**

1. Does the alignment of attention in the paper require the introduction of some prior knowledge?

2. Do distributional alignment[1, 2] and this type of representation alignment conflict with each other? Have you conducted further experiments on combining them?

3. From the description in the paper, the representation alignment of FilpClass seems to be distribution-agnostic. Can its performance still be maintained under long-tail distribution conditions[3]?

[1] Open-World Semi-Supervised Learning. ICLR 2022.
[2] Robust Semi-Supervised Learning when Not All Classes have Labels. NeurIPS 2022.
[3] Towards Distribution-Agnostic Generalized Category Discovery. NeurIPS 2023.

---

> ### Author Rebuttal · Authors · 2024-08-07
>
> We appreciate your meticulous review and valuable feedback.
>
> **Response to W1**
> > ***The content regarding "the Hopfield Network" and the underlying motivation could be enhanced.***
>
> Thank you for the suggestions! We have moved the **state-query capacity of the Hopfield Network** from Appendix A.1 to the main body and added more related work as suggested by ***Reviewer MRfK***.
>
> We clarified the motivation by showing how the Hopfield Network's pattern storage and retrieval align with our goal of attention alignment. Its dynamic updates and energy minimization inspire our method to adjust the teacher's attention based on the student's focus, ensuring better synchronization and learning outcomes.
>
> &nbsp;&nbsp;
>
> **Response to W2**
> > ***SPTNet [1] should also be compared in the paper.***
>
> Thanks for the suggestion. Below, we compare *FlipClass* with *SPTNet* [1] and report the accuracy of Old, New, and All classes.
>
> 1. Superiority: Our method achieves the best results across all datasets, as shown in **Table 3 in the attached PDF**.
> 2. Comparison: While both *SPTNet* and *FlipClass* target Open-World semi-supervised learning, *SPTNet* focuses on **information extraction at the pixel level**.
> In contrast, *FlipClass* emphasizes **matching supervision signals at the representation level**.
>
> &nbsp;&nbsp;
>
> **Response to W3**
> > ***Differences between distribution alignment [2,3] and representation alignment.***
>
> Thank you for this insightful comment.\
> Distribution alignment in [2,3] utilizes KL divergence to regularize the predicted probabilities to be **close to a prior probability distribution $P$ to better learn new classes**.
>
> In comparison, our representation alignment **does not require this prior $P$ of classes**.
> Additionally, our approach focuses on aligning the learned representations between teacher and student, **improving consistency without relying on a predefined class distribution**.
> This makes our method more adaptable and effective in diverse scenarios.
>
> ---
> \
> **Response to Q1**
> > ***Does the representation alignment require prior knowledge?***
>
> No, it does not. **Unlike [2,3], our method inherits from the Teacher-Student framework, where the teacher provides higher quality supervision signals,
> forming a weak prior to guide the student's learning (i.e., $p(student∣teacher)$)**.
>
> Since this weak prior can have flaws and may not reflect the true distribution accurately, *FlipClass* further formulates
> $p(teacher∣student)$ to enable iterative mutual learning between the teacher and student.
> This approach helps avoid incorrect guidance stemming from the teacher's overconfidence in its weak prior.
>
> &nbsp;&nbsp;
>
> **Response to Q2**
> > ***Can distributional alignment [2,3] be combined with FlipClass's representation alignment?***
>
> Interesting question! Yes, our representation alignment can be incorporated with other distribution alignment strategies.
>
> The distribution alignment can be applied by adjusting the parametric classification loss defined in Sec 4.2 L198-203, seamlessly collaborating with our representation alignment in Sec 4.1.\
> We conducted experiments on *Stanford Cars*, by combining these distribution alignment strategies [2,3,4].
> **The results are as provided in Table 4 in the attached PDF**.
>
> The reasons why directly combining these distribution alignment strategies does not bring improvement are:
> 1. ***ReMixMatch*** [4]: \
> ReMixMatch maintains a running average of the model's predictions on unlabeled data, $\tilde{p}(y)$, which scales the model prediction $q$ by the ratio $p(y)/ \tilde{p}(y)$, forming a valid probability distribution:
> ​$\tilde{q} =\text{Normalize}(q×p(y)/ \tilde{p}(y))$. \
> However, during the early training stage **in Open-World scenarios, the model's predictions are inaccurate on new classes, prone to making incorrect predictions of prior distributions
> ($\tilde{p}(y)$) on unlabeled data**.
> 2. ***Prior Distribution Alignment*** [2,3]: \
> This approach regularizes the model with maximum entropy regularization using an even-distributed prior distribution.
> This works on *Cifar* and *ImageNet*, which have balanced class distributions. \
> However, **the *SSB Bench*** (*e.x.*, *Stanford Cars*) **has unbalanced class distributions (shown in Fig. 15 in Appendix C.3)**.
> Without prior knowledge of the class distribution (especially on new classes), applying this naive distribution alignment degrades performance on both new and old classes.
>
> Further exploration of distribution alignment on unbalanced datasets is an interesting direction, and we'll work on this to combine it with representation alignment.
>
> &nbsp;&nbsp;
>
> **Response to Q3**
> > ***Performance under long-tail distribution conditions [5]?***
>
> Thank you for the suggestion to validate our method under long-tail distribution conditions. \
> Yes, *FlipClass*'s representation alignment is indeed **distribution-agnostic**.\
> **We conducted experiments following the setting of [5]. The results are shown in Table 5 in the attached PDF.**
>
> While *FlipClass* does not surpass BaCon [5], it still performs well.\
> Unlike non-parametric methods (*GCD, OpenCon, BaCon*), *FlipClass* and *SimGCD* reduce clustering time costs.
> Although non-parametric methods are often more robust, *FlipClass*'s attention alignment strategy demonstrates effective, distribution-agnostic performance.
>
> &nbsp;&nbsp;
>
> **We've included above explanation and extended experiments in the revised manuscript. Once again, thanks for your constructive suggestions.**
>
> &nbsp;&nbsp;
>
> *[1] SPTNet: An efficient alternative framework for generalized category discovery with spatial prompt tuning. ICLR 2024.*
>
> *[2] Open-World Semi-Supervised Learning. ICLR 2022.*
>
> *[3] Robust Semi-Supervised Learning when Not All Classes have Labels. NeurIPS 2022.*
>
> *[4] Remixmatch: Semi-supervised learning with distribution alignment and augmentation anchoring. ICLR. 2020.*
>
> *[5] Towards Distribution-Agnostic Generalized Category Discovery. NeurIPS 2023.*

---

> > ### Comment · Area_Chair_F1iG · 2024-08-07
> >
> > Thank you to the authors for this response. Dear reviewer P1mR: Could you check whether the authors addressed your points? It would be particularly helpful to receive your update.

---

> > > ### Comment · Reviewer_P1mR · 2024-08-08
> > >
> > > The authors have addressed my concerns. Thanks AC and authors.

---

> > > > ### Comment · Area_Chair_F1iG · 2024-08-08
> > > >
> > > > Thank you very much for your response!

---

> > ### Comment · Reviewer_P1mR · 2024-08-08
> >
> > Thanks for your good responses! And you have addressed my concerns. I will increase my score to 7, this is a comprehensive paper.

---

> > > ### Author Response · Authors · 2024-08-08
> > > **Further Reply to Reviewer P1mR**
> > >
> > > Dear Reviewer P1mR,
> > >
> > > We greatly appreciate your satisfaction with our responses, and very glad you increase the rating! We will make comprehensive revisions to our work based on your comments in order to further improve the quality of our work.
> > >
> > > Thanks again for your valuable suggestions and comments. We enjoy communicating with you and appreciate your efforts!

---

### Official Review · Reviewer_MRfK · 2024-07-12

**Soundness:** 3
**Presentation:** 4
**Contribution:** 3
**Rating:** 6
**Confidence:** 4

**Summary:**

This work proposes an attention alignment technique based on the Hopfield network energy function. Specifically, this work proposes to update the teacher model to increase the posterior teacher likelihood given the current student, which is modeled with the Hopfield network energy-based model. The teacher update process derived from the conditional score function is proven to be globally convergent. Experimental results showcase its considerable improvements over previous state-of-the-art baselines. The attention alignment technique is well-motivated with in-depth analysis. The contributions of this work include (1) investigating and discovering the attention alignment inconsistency between student and teacher models for the generalized category discovery problem, (2) proposing a theoretically inspired attention alignment method to address this issue, which is guaranteed to converge globally, and (3) achieving considerable performance gains compared with previous sota models.

**Strengths:**

strengths:

- The idea of updating the teacher by aligning attention layers of the student is innovative.
- The experimental results are comprehensive.
- Different alignment strategies have been compared and the superior efficacy of attention alignment has been validated.
- The global convergence of the update rule of the teacher model is proved.

**Weaknesses:**

weaknesses:

- Writing: It is highly recommended to put the related work in the main content rather than in the appendix.
- Ablations: There lacks the analysis on the hyperparameter \alpha, and the rationale behind setting \alpha=0 is to be explained.
- line 141: incorrect symbol
- Ablations: Better to provide comparisons in the ablations on varying class numbers.

**Questions:**

questions:

- Baselines: Some baselines on generic and fine-grained datasets are not consistent in Tables 1 & 2, could you please tell us the reasons? It is better to make them consistent unless there is any special reason.
- References: Is there any related work on utilizing the attention alignment technique?
- Figures: Are the two methods visualized in figure 8b within the same projection space and with the same scale? Better provide zoom-in comparisons since there are two many classes.

**Limitations:**

Social impact: current methods are not applicable to real-world safety-demanding applications.

---

> ### Author Rebuttal · Authors · 2024-08-07
>
> We appreciate your valuable feedback. Before addressing your inquiries, we wish to clarify certain weaknesses highlighted in the review that we believe require further elucidation.
>
> **Response to W1**
> > ***Put the related work in the main content rather than in the appendix.***
>
> Thanks for your suggestions!
>
> We've incorporated a concise version of related work in appendix into the revised main content, and make the experimental analysis more concise while keep it easy to follow.
>
> &nbsp;&nbsp;
>
> **Response to W2**
> > ***Analysis on the hyperparameter \alpha, and the rationale behind setting \alpha=0.***
>
> Thank you for pointing this out.
>
> **Recap of Eq. 8**: the regularization term introduced by the prior energy prevents a single teacher key from dominating the attention, while the attention term updates the teacher keys in the direction of student queries, improving the consistency of focused patterns.
>
> In practice, we found that a **nonzero** $\alpha$ **(in Eq. 8) often leads to over-penalization, causing some teacher keys to vanish, and we found that setting $\alpha=0$ gives the most consistent results**.
>
> During experiments, we conducted an **analysis of $\alpha$ on the Stanford Cars dataset** to determine the optimal value and applied it to all datasets.
>
> **The results are provided in Table 1 in the attached PDF**.
>
> These results highlight that $\alpha=0$ yields the best performance across all categories.
>
> We hope this clarifies our rationale.
>
> &nbsp;&nbsp;
>
> **Response to W3**
> > ***line 141: incorrect symbol.***
>
> We sincerely appreciate your meticulous review, and we have corrected the mentioned typo in our revised manuscript.
>
>
> &nbsp;&nbsp;
>
> **Response to W4**
> > ***Provide comparisons in the ablations on varying class numbers.***
>
> Yes, we agree.
>
> We have included ablations on varying class numbers in **Appendix C.4 "Robustness to Number of Classes"**. Please kindly refer to it.
>
> We will also consider incorporating these results into the main content.
>
> In summary, our model demonstrates robust performance across different numbers of classes, maintaining consistent accuracy and stability.
>
> ---
> \
> **Response to Q1**
> > ***Some baselines on generic and fine-grained datasets are not consistent in Tables 1 & 2, could you please tell us the reasons?***
>
> Sure, I'd like to, and thanks for your meticulous review.
>
> The reasons for the inconsistency in baselines between Tables 1 and 2 are as follows:
> 1. Inconsistency between Tables 1 and 2: **In detail, *XCon*, *PCAL*, *$\mu$GCD* [1] are missed in Table 2, which are present in Table 1**.
> 2. **Page Limits**: Due to space constraints, we removed some methods (*e.g., XCon*, *PCAL*) from Table 2 on generic datasets, since this dataset is less challenging compared to fine-grained datasets. \
> **The results for XCon and PCAL on both generic and fine-grained datasets are provided in Table 2 in the attached PDF**.
> 3. **Unavailable Code**: $\mu$GCD [1] did not release their code, and their official paper does not report results on generic datasets. We attempted to reproduce their results but were unsuccessful.
> Therefore, we only reported their results on fine-grained datasets.
> 4. **Method Ordering**: We **sorted the methods based on their All Acc**, which may result in different orders between Tables 1 and 2.
>
> &nbsp;&nbsp;
>
> **Response to Q2**
> > ***Is there any related work on utilizing the attention alignment technique?***
>
> Yes, and thanks for the suggestions. Here is a concise summary of the attention alignment strategies in related works:
> - *RDAN* [2]: Utilizes a dual attention network to infer visual-semantic alignments by aligning attention across image regions and textual descriptions.
> - *MAL* [3]: Employs multi-attention localization to discover discriminative parts of objects for zero-shot learning, aligning attention based on semantic guidance.
> - *Alignment Attention* [4]: Focuses on aligning key and query distributions to improve the accuracy and effectiveness of attention mechanisms.
> - *Multi-level Representation Learning* [5]: Uses semantic alignment to enhance multi-level representation learning, aligning attention across different levels of representation.
>
> Differences and Contributions of *FlipClass*:
> - **Generalized Category Discovery**: FlipClass is tailored for generalized category discovery in open-world settings, addressing challenges in semi-supervised learning scenarios and maintaining robust performance across different datasets and distributions.
> - **Teacher-Student Framework**: Unlike other methods that focus on visual-semantic or multi-level semantic alignments, FlipClass integrates attention alignment within a Teacher-Student framework.
> This allows the teacher's guidance to adapt based on the student's learning, promoting better synchronization and performance.
>
> We also provided experimental comparison with MAL [3] in **Response to Q1** with ***Reviewer MqmB***, please kindly refer to it.
>
> &nbsp;&nbsp;
>
> **Response to Q3**
> > ***Are the two methods visualized in figure 8b within the same projection space and with the same scale?***
>
> Yes, during visualization. We utilize the same components for t-SNE and PCA to ensure the results of compared methods are projected in the same space with the same scale.
>
> > ***Better provide zoom-in comparisons.***
>
> We appreciate your advice, and please kindly refer to **Fig.1 in the attached PDF**, which present the zoom-in comparison for better analysis of the representation space.
>
> &nbsp;&nbsp;
>
> *[1] μGCD: No representation rules them all in category discovery. NeurIPS, 2024.*
>
> *[2] Multi-level visual-semantic alignments with relation-wise dual attention network for image and text matching. IJCAI. 2019.*
>
> *[3] Semantic-guided multi-attention localization for zero-shot learning. NeurIPS. 2019.*
>
> *[4] Alignment attention by matching key and query distributions. NeurIPS. 2021.*
>
> *[5] Multi-level representation learning with semantic alignment for referring video object segmentation. CVPR. 2022.*

---

> > ### Comment · Area_Chair_F1iG · 2024-08-07
> >
> > Thank you to the authors for this response. Dear reviewer MRfK: Could you check whether the authors addressed your points? It would be particularly helpful to receive your update.

---

> ### Author Response · Authors · 2024-08-13
>
> Dear Reviewer MRfK,
>
> Thank you again for your time and insightful comments! We have comprehensively revised our work according to your comments (please kindly refer to the rebuttal above). We hope we have addressed your concerns regarding the **analysis and explanation of the role of $\alpha$ in Eq. (8)**, **compared methods in Tables 1&2** and **related work on attention alignment**, *etc*.
>
> **Since the discussion is about to close, we would be grateful if you would kindly let us know of any other concerns and if we could further assist in clarifying any other issues.**
>
> Thanks a lot again, and with sincerest best wishes.
>
> Authors

---

> > ### Comment · Reviewer_MRfK · 2024-08-13
> >
> > I appreciate the authors' great effort in providing further details. My primary concerns on related work and $\alpha$ has been addressed. I hope the author could add these related works on attention alignment to the final script. I would like to increase the score by one.

---

> > > ### Author Response · Authors · 2024-08-13
> > > **Thanks for Your Further Response**
> > >
> > > Dear Reviewer MRfK,
> > >
> > > We greatly appreciate your helpful comments and your satisfaction with our responses! We will add these related works on attention alignment to our revised manuscript, and make comprehensive revisions based on the above important discussions and highlight them.
> > >
> > > Thanks again for your valuable suggestions and comments. We really enjoy communicating with you and appreciate your efforts.

---

### Official Review · Reviewer_MqmB · 2024-07-17

**Soundness:** 3
**Presentation:** 4
**Contribution:** 3
**Rating:** 7
**Confidence:** 3

**Summary:**

The paper introduces FlipClass, a dynamic teacher-student attention alignment strategy designed to address the challenges of Generalized Category Discovery (GCD) in open-world scenarios. Unlike traditional teacher-student frameworks, FlipClass updates the teacher’s attention to align with the student’s evolving focus, thereby promoting consistent pattern recognition and synchronized learning across both old and new classes. Extensive experiments validate FlipClass’s superiority over existing methods, establishing new benchmarks in GCD performance.

**Strengths:**

1.	Innovative Approach:
–	The dynamic teacher-student attention alignment strategy is novel in continuously updating the teacher's focus based on the students, ensuring synchronized learning and significantly advancing over static models.
2.	Interpretability:
–	Clear visualizations, including attention heatmaps and energy dynamics, effectively demonstrate how attention alignment between teacher and student improves learning outcomes.
3.	Training Details:
–	The detailed description of the teacher-attention update rule and the theoretical foundation provided in the appendices contribute to a thorough understanding of the training process.
4.	Experimental Validation:
–	The extensive experiments conducted on various benchmarks, strongly support the claims made by the authors; the experimental analysis clearly demonstrates the importance of attention alignment, further validating the approach.

**Weaknesses:**

1.	Experiments:
–	Certain details in the ablation study, like the impact of strong augmentations and regularization, could be more clearly explained.
–	The font size in Fig. 6 is too small and should be increased for better readability.
2.	Writing and Presentation:
–	Consistency in terminology and notation throughout the paper needs improvement to avoid confusion. For example, clearly distinguishing between different types of augmentations and regularization techniques used in the experiments.

**Questions:**

1.	While the paper claims that FlipClass significantly improves attention alignment (a design of $\R$ in Insight 3.1), could you provide more detailed quantitative metrics and comparative analyses with other attention alignment strategies to strengthen this claim?
2.	In Table 4, why does the accuracy of new classes increase significantly at the expense of the accuracy of old classes on the CUB dataset? Can you explain this phenomenon?

**Limitations:**

The necessary experiment of ablation study misses explanations for the strong augmentation and regularization.

---

> ### Author Rebuttal · Authors · 2024-08-07
>
> We appreciate your positive feedback and constructive comments.  Before addressing your inquiries, we wish to clarify certain weaknesses highlighted in the review that we believe require further elucidation.
>
>
> **Response to W1**:
> >  ***Certain details in the ablation study, like the impact of strong augmentations and regularization, could be more clearly explained.***
>
> Thank you for your suggestion.
>
> The impact of strong augmentations and regularization are as follows:
> - **Strong augmentations** are introduced to expose the student network to a **wider range of image variations**, thereby **enhancing its robustness and generalization capabilities**.
> - **Regularization** during the attention update **integrates the prior energy of the teacher**, preventing any single student pattern from overly influencing the teacher’s attention.
>
> These clarifications have been added to Section 4 in the revised manuscript.
>
>
> &nbsp;&nbsp;
>
> **Response to W2**:
> >  ***Writing and Presentation: Consistency in terminology and notation throughout the paper....***
>
> Thanks for your detailed review. We've revised these notations and fonts to improve the readability of our paper.
>
> ---
>
> &nbsp;&nbsp;
>
> **Response to Q1**:
> >  ***While the paper claims that FlipClass significantly improves attention alignment (a design of $\color{red}\Re$ in Insight 3.1), could you provide more detailed quantitative metrics and comparative analyses with other attention alignment strategies to strengthen this claim?***
>
> Certainly! Thank you for the suggestion.
>
> Thank you for your advice. Below, we provide detailed **quantitative comparison of different alignment strategies between the student and teacher attention maps**.
>
> The strategies we experimented with include *$l_2$ loss, Kullback–Leibler divergence (KLD) loss, and Correlation Alignment (CORAL) loss*, as well as the *Semantic-Guided Multi-Attention Alignment (MAL)* method [1].
>
> | Attention Alignment Methods | CUB  |      |      | SCars |      |      |
> | --------------------------- | ------ | ------ | ------ | ------- | ------ | ------ |
> |                             | All  | Old  | New  | All   | Old  | New  |
> | **Ours (*FlipClass*)**        | **71.3** | **71.3** | **71.3** | **63.1**  | **81.7** | **53.8** |
> | $l_2$ Loss                  | 62.1 | 63.6 | 61.4 | 48.2  | 64.0 | 40.3 |
> | KLD Loss                    | 64.5 | 70.3 | 61.7 | 52.7  | 72.8 | 42.6 |
> | CORAL                       | 61.1 | 67.7 | 57.8 | 48.3  | 67.9 | 38.5 |
> | MLA                         | 68.3 | 70.4 | 67.2 | 56.9  | 73.4 | 48.7 |
>
>
> Our method achieves the highest accuracy across both datasets, demonstrating the effectiveness of our energy-based alignment strategy.
> This approach allows for dynamic and nuanced adjustment of the teacher's attention, leading to better alignment with the student's evolving synchronized learning.
>
> While MAL shows closer performance to our method compared to other strategies, it still does not match our accuracy, underscoring the unique advantages of our energy-based strategy.
>
>
> &nbsp;&nbsp;
>
> **Response to Q2**:
> >  ***In Table 1, why does the accuracy of new classes increase significantly at the expense of the accuracy of old classes on the CUB dataset?***
>
> Thank you for your thorough review.
>
> As shown in Table 1, methods such as *PCAL, $\mu$GCD*, and *AdaptGCD* also achieve comparatively similar accuracy of new and old classes on the CUB dataset.
>
> We attribute this to the small scale of the CUB dataset, which contains only 6,000 images with a large class split (200).
> This smaller dataset size might reduce the tendency to overfit the old classes, leading to a more balanced accuracy across new and old classes.
>
> &nbsp;&nbsp;
>
> *[1] Semantic-guided multi-attention localization for zero-shot learning. NeurIPS. 2019.*

---

> ### Author Response · Authors · 2024-08-13
>
> Dear Reviewer MqmB,
>
> Thank you again for reviewing our manuscript. We have tried our best to address your questions (see our rebuttal in the top-level comment and above), and revised our paper by following suggestions from all reviewers.
>
> Please kindly let us know if you have any follow-up questions or areas needing further clarification. Your insights are valuable to us, and we stand ready to provide any additional information that could be helpful.

---

### Author Rebuttal · Authors · 2024-08-07

## **General Response to All Reviewers**

We sincerely thank all reviewers for the time they spent reviewing our manuscript and for their thoughtful feedback. We appreciate that the reviewers found our paper theoretically and methodologically novel, with strengths such as:
- the **idea** of dynamic teacher-student attention alignment strategy to be **innovative and interesting** (***Reviewers MqmB, MRfK, P1mR***);
- our proposed method to be **well-organized, clearly written**, and providing **considerable contributions** to the field (***Reviewers Btrz, P1mR***);
- and overall our **analysis to be comprehensive**, with **detailed experimental validation and theoretical insights** (***Reviewers MqmB, MRfK, Btrz***).

The attached PDF includes (1) **the analysis of hyperparameter $\alpha$**, (2) **zoom-in cluster visualization** (***Reviewer MRfK***), (3) **extended comparison with SPTNet**,  (4) **combinations with distribution alignment strategies**, and (5) **results on long-tailed distribution conditions** (***Reviewer MRfK***).

We have provided detailed responses to individual reviewers below, and have included additional experiments suggested by the reviewers in the Author Response PDF.

We are also pleased to publicly release all code.

Please let us know if you have any additional questions or concerns. We are happy to provide clarification.

---

### Decision · Program_Chairs · 2024-09-25

**Decision:**

Accept (oral)

**Comment:**

All reviewers agree that the presented work is innovative, of broad interest to the NeurIPS community, and well-evaluated experimentally (although some questions had to be cleared up during the discussion phase). Given the clear consensus on the quality of this paper, acceptance is warranted. Authors are advised to include the clarifications that emerged during the discussion phase in their camera-ready version.